# A rigorous approach to the specific surface area evolution in snow during temperature gradient metamorphism

Anna Braun[1,2], Kévin Fourteau[1], and Henning Löwe[1]

[1]Group Snow Physics, Research Unit Snow and Atmosphere, WSL Institute for Snow and Avalanche Research SLF, 7260 Davos Dorf, Switzerland
[2]Laboratory of Cryospheric Sciences, School of Architecture, Civil and Environmental Engineering, Ecole Polytechnique Federale de Lausanne, 1015 Lausanne, Switzerland

**Correspondence:** Henning Löwe (loewe@slf.ch)

**Abstract.** Despite being one of the most fundamental microstructural parameters of snow, the specific surface area (SSA) dynamics during temperature gradient metamorphism (TGM) have so far been addressed only within empirical modeling. To surpass this limitation, we propose a rigorous modeling of SSA dynamics using an exact equation for the temporal evolution of the surface area, fed by pore-scale finite element simulations of the water vapor field coupled with the temperature field on X-ray computed tomography images. The proposed methodology derives from physics' first principles and thus does not rely on any empirical parameter. Since the calculated evolution of the SSA is highly sensitive to fluctuations in the experimental data, we quantify the impact of these fluctuations within a stochastic error model. In our simulations, the only poorly constrained physical parameter is the condensation coefficient $\alpha$. We address this problem by simulating the SSA evolution for a wide range of $\alpha$ and estimate optimal values by minimizing the differences between simulations and experiments. This methodology suggests that $\alpha$ lies in the intermediate range $10^{-3} < \alpha < 10^{-1}$ and slightly varies between experiments. Also, our results suggest a transition of the value of $\alpha$ in one TGM experiment, which can be explained by a transition in the underlying surface morphology. Overall, we are able to reproduce very subtle variations in the SSA evolution with correlations of $R^2 = 0.95$ and 0.99, respectively, for the two considered TGM time series. Finally, our work highlights the necessity of including kinetics effects and of using realistic microstructures to comprehend the evolution of SSA during TGM.

## 1 Introduction

The specific surface area (SSA) of snow is the interface area between ice and air in the microstructure of porous snow, normalized per volume. The SSA is a crucial parameter for the optical albedo of snow (Dumont et al., 2014), fluid permeability (Zermatten et al., 2014), avalanche prediction (Schweizer et al., 2003), microwave remote sensing (Picard et al., 2022), or chemical exchange with the atmosphere (Hanot and Dominé, 1999). The SSA evolution in time is one key to quantifying metamorphism (Legagneux et al., 2004; Domine et al., 2006; Pinzer et al., 2012; Wang and Baker, 2014; Harris Stuart et al., 2023) and needs to be faithfully parameterized in snow cover models to capture the evolution of physical properties. Temperature gradient metamorphism (TGM) is by far the most important type of metamorphism in dry, natural snow covers (Schneebeli

and Sokratov, 2004; Legagneux et al., 2004), since gradient-free (i.e., isothermal) conditions exist at most in deep polar firn. However, a detailed physical understanding of the SSA evolution under TGM is still lacking.

Detailed experimental data on TGM can be conveniently acquired nowadays through X-ray micro-computed tomography ($\mu$CT). Imaging of snow samples with $\mu$CT was developed over the last two decades (Coleou et al., 2001; Flin et al., 2004; Schneebeli and Sokratov, 2004; Schleef and Loewe, 2013) and provides 3D insight into the microstructure that is otherwise invisible to the naked eye. In contrast to many destructive snow measurement methods, $\mu$CT preserves the structure of the snow. Since the entire snow microstructure is available, any parameter of interest, especially SSA, can be computed within well-characterized uncertainties due to reconstruction and image analysis (Hagenmuller et al., 2016). By using instrumented sample holders to constrain temperatures and temperature gradients, in-situ time-lapse observations of the microstructure during TGM are obtained (Kaempfer et al., 2005; Pinzer et al., 2012; Calonne et al., 2014a; Hammonds et al., 2015; Wiese and Schneebeli, 2017; Li and Baker, 2022). While many SSA evolution curves originated from these studies, none of them has been convincingly reproduced from a physical model.

Physical models of snow metamorphism must comply with the ice crystal growth dynamics at the pore scale (Krol and Löwe, 2016), which includes heat and mass diffusion, accommodated by attachment kinetics controlling the deposition and sublimation of water molecules onto the ice lattice (Colbeck, 1983; Libbrecht, 2005). Secondary effects on the temporal SSA evolution might be expected from other processes like mechanical deformation (Wang and Baker, 2013; Schleef et al., 2014), advection of air in the porosity (Ebner et al., 2016; Jafari et al., 2022). In this picture, one key parameter driving snow metamorphism is the condensation coefficient $\alpha$, also called attachment, kinetic or sticking coefficient (Libbrecht, 2005; Kaempfer and Plapp, 2009; Krol and Löwe, 2016; Demange et al., 2017; Fourteau et al., 2021b; Bouvet et al., 2022) that controls the kinetics of vapor deposition and sublimation. The condensation coefficient is applicable at the micro-meter scale of ambient diffusion processes and thereby subsumes the underlying nano-scale kinetics resulting from the molecular dynamics on the surface of the ice crystal lattice (Saito, 1996). Many measurement and modeling attempts carefully characterize $\alpha$ for ice crystals (Libbrecht, 2005; Hobbs, 2010; Barrett et al., 2012; Libbrecht and Rickerby, 2013; Pokrifka et al., 2020). Nevertheless, $\alpha$ is experimentally challenging to constrain even for isolated crystal growth. One reason is the fundamental, experimental difficulty of inverting growth data as soon as diffusion is involved (Libbrecht, 2005). The other reason is that $\alpha$ depends on numerous effects such as temperature, supersaturation, and crystallographic orientation (Saito, 1996; Libbrecht, 2005). The large variations between basal and prismatic surface kinetics are, for example, the key to snow crystal morphology (Barrett et al., 2012). The situation is even more complicated in the snow cover where many different surface orientations exist simultaneously (Granger et al., 2021). Therefore, the kinetics is more difficult to assess in snow, and only a few studies exist constraining $\alpha$ from the comparison of $\mu$CT-based simulations with experiments (Bouvet et al., 2022; Fourteau et al., 2021a). Thus, $\alpha$ constitutes the great unknown in snow metamorphism as commonly stressed in TGM models (Miller and Adams, 2009; Kaempfer and Plapp, 2009; Calonne et al., 2014b).

Model attempts characterizing TGM can be classified by their treatment of attachment kinetics and whether the microstructure is taken from $\mu$CT or geometrically idealized. Using $\mu$CT images, (Flin and Brzoska, 2008) calculated deposition fluxes in the absence of kinetics under the assumption of local equilibrium at the interface (diffusion-limited growth). A similar ap-

proximation was used in (Krol and Löwe, 2016) to relate the temperature-gradient-driven deposition fluxes to measured, local interface velocities. The latter can be considered as a generalization of the (diffusion-limited) air bubble migration under a tem-

perature gradient in ice (Shreve, 1967) to complex geometries. However, the assumption of purely diffusion-limited growth was already questioned (Krol and Löwe, 2018) due to contradictions with the measured SSA evolution. The $\mu$CT-based theoretical homogenization (Calonne et al., 2014b), in contrast, applies to the slow kinetics (i.e., kinetics-limited) regime. The intermediate regime from diffusion to kinetics vapor transport under a temperature gradient was numerically analyzed in (Fourteau et al., 2021a), where the latter approach is physically similar to the phase field model (Kaempfer and Plapp, 2009). Since the

choice of $\alpha$ has a significant impact on numerical effort, it is not surprising that the majority of modeling attempts exist for simplified geometries (mostly spheres) (Adams and Brown, 1982; Colbeck, 1983; Albert and McGilvary, 1992; Miller and Adams, 2009), at the expense of microstructural realism. The most widely used models for predicting the SSA evolution under TGM are those implemented in snow cover models e.g., (Flanner and Zender, 2006). Like other simplified models, (Flanner and Zender, 2006) neglect kinetics and employ diffusion-limited growth for the distribution of spherical particles. Due to

the involved empirical parameters (mean sphere radius and spacing), which prevent an unambiguous mapping onto arbitrary microstructures, validating these models through $\mu$CT laboratory experiments would remain inconclusive.

In principle, no empiricism is required, and the SSA evolution for arbitrary 3D microstructure can be computed exactly (Krol and Löwe, 2018), as long as the required parameters are supplied. The surface area equation is rigorously formulated on the basis of a growth rate that can be computed from the interfacial curvature and the interface velocity $v_{\mathbf{n}}$ after surface

averaging. While the interfacial curvature is a purely geometrical quantity that can directly be computed from a $\mu$CT image, $v_{\mathbf{n}}$ is a physical quantity that further depends on the involved physical processes. In this framework, any model that predicts $v_{\mathbf{n}}$ as the result of 3D heat and mass diffusion with interface kinetics could be employed here, either phase field models (Kaempfer and Plapp, 2009) or diffusion models (Fourteau et al., 2021b). Both are equivalent in view of the involved physics and only differ in their representation of the interface. This route to the SSA evolution in TGM is rigorous (apart from numerical

approximations) but has never been pursued before. Advancing on this route is the aim of the present work. To this end, we combine a finite element (FE) solution of the pore-scale heat and mass diffusion equations following (Fourteau et al., 2021b) with the exact surface area equation from (Krol and Löwe, 2018) in order to reproduce the SSA evolution during TGM from the four-dimensional (4D) $\mu$CT image data from (Pinzer et al., 2012).

The manuscript is organized as follows. The theoretical background for pore-scale diffusion and the SSA is presented in

Sect. 2. In Sect. 3, we describe the numerical procedures (meshing, FE solution, image processing), a simple stochastic error analysis, and the validation of our numerical workflow against an analytical solution. The simulations for the TGM time series are shown in Sect. 4 and discussed in Sect. 5.

## 2 Theoretical background

### 2.1 Heat and vapor transfer at the pore scale

For an arbitrary snow structure, morphological changes during metamorphism are predominantly driven by the coupled diffusion of heat and mass together with ice-air interface evolution due to deposition and sublimation of vapor. In the following, we closely follow the descriptions by (Kaempfer and Plapp, 2009; Calonne et al., 2014b; Krol and Löwe, 2016; Fourteau et al., 2021a). We consider a representative snow volume at the micro-scale consisting of ice and air and denote the sub-domains occupied by the ice and air phases by $\Omega_i$ and $\Omega_a$, respectively. In the following, subscripts $i$ and $a$ denote quantities which are defined in the respective domains $\Omega_i$ and $\Omega_a$. Due to the separation of time scales between heat and mass diffusion in the pores and the evolution of the interface, we employ the common assumption of small particle Péclet number (Libbrecht, 2005) and consider stationary heat and mass diffusion equations (i.e., Laplace equations). Furthermore, we neglect the influence of mechanical deformation, as usually done in pore-scale metamorphism models (e.g., (Calonne et al., 2014b; Krol and Löwe, 2016)). We also neglect the potential presence of convection and air advection in the pore space. These assumptions are consistent with the experimental data used in this article, obtained under controlled laboratory conditions (Pinzer et al., 2012). They are also good candidates in terms of minimum-required complexity to model SSA evolution from pore-scale physics. The partial density of water vapor in air $\rho_v$ and the ice and air temperatures $T_i$ and $T_a$, respectively, are governed by

$$D_v \nabla^2 \rho_v = 0 \qquad\qquad \text{in } \Omega_a \tag{1}$$

$$\kappa_a \nabla^2 T_a = 0 \qquad\qquad \text{in } \Omega_a \tag{2}$$

$$\kappa_i \nabla^2 T_i = 0 \qquad\qquad \text{in } \Omega_i \tag{3}$$

where $D_v$ is the vapor diffusion constant in air, $\kappa_i$ and $\kappa_a$ are the thermal diffusivities of ice and air, respectively.

The heat and mass diffusion equations are coupled via boundary conditions on the ice-air interface $\Gamma$. The mass conservation at the ice-air interface is linked to the water vapor concentration by a Stefan-type condition

$$(\rho_i - \rho_v)\, v_\mathbf{n} \;=\; D_v\, \mathbf{n} \cdot \nabla \rho_v \qquad\qquad \text{on } \Gamma \tag{4}$$

where $\rho_i$ denotes the ice density and $\mathbf{n}$ the unit normal vector field on $\Gamma$ which is oriented into the pore space $\Omega_a$ and $v_\mathbf{n}$ is the interface velocity on $\Gamma$ in the direction of $\mathbf{n}$. The velocity $v_\mathbf{n}$ is therefore positive for deposition and negative for sublimation.

The conservation of energy requires the continuity of temperature and heat flux on the ice-air interface according to

$$T_i \;=\; T_a \qquad\qquad \text{on } \Gamma \tag{5}$$

$$\kappa_i\, \mathbf{n} \cdot \nabla T_i \;=\; \kappa_a\, \mathbf{n} \cdot \nabla T_a \qquad\qquad \text{on } \Gamma \tag{6}$$

As by (Krol and Löwe, 2016), the latent heat during the sublimation and deposition is neglected for reduced model complexity. Since mass and energy conservation involves the unknown interface velocity $v_\mathbf{n}$, the internal boundary conditions must be completed by a constitutive law that characterizes $v_\mathbf{n}$ during crystal growth. Here, we employ the Hertz-Knudsen law (Libbrecht,

2005; Kaempfer and Plapp, 2009; Fourteau et al., 2021a), which includes the impact of interfacial curvature on the equilibrium vapor concentration (Gibbs-Thomson effect) according to

$$\rho_v = \rho_{v,s}(T)(1 + d_0\,H) + \frac{\rho_i}{\alpha\,v_{\text{kin}}}\,v_\mathbf{n} \qquad\qquad \text{on } \Gamma \tag{7}$$

The equilibrium (or saturation) vapor concentration on a flat surface at temperature $T$ is denoted by $\rho_{v,s}(T)$, the capillary length by $d_0$, the mean curvature by $H$, the condensation coefficient by $\alpha$ and the kinetic velocity by $v_{\text{kin}}$. The capillary length is related to $d_0 = \gamma\,a^3/(k_B\,T)$, where $\gamma$ is the interfacial free energy, $a$ is the mean intermolecular spacing of water molecules in ice and $k_B$ is the Boltzmann constant. The kinetic velocity is defined here as $v_{\text{kin}} = \sqrt{k_B\,T/(2\,\pi\,m)}$ with the mass of water molecule $m$. This definition follows (Fourteau et al., 2021a) and thus differs from the definition in (Libbrecht, 2005). In the Hertz-Knudsen equation, the condensation coefficient $\alpha$ is defined as the probability of a water molecule sticking to a surface after impinging on it. Therefore, values in the range $[0,1]$ are commonly desired, where $\alpha \to 0$ corresponds to slow surface kinetics and for $\alpha \approx 1$ the diffusion-dominated regime will be attained (Libbrecht, 2005; Fourteau et al., 2021a). Mathematically the equation remains well-defined also for $\alpha > 1$, which may be physically interpreted as deviations from the local constitutive behavior (Eq. (7)) due to non-local surface processes (Libbrecht, 2005). Although $\alpha$ is known to depend on temperature, supersaturation, crystallographic orientation and vary on different parts of the ice-air interface (Libbrecht, 2005), we rely on the simplifying assumption of a single and constant $\alpha$ value. It should thus rather be understood as an *effective* condensation coefficient.

## 2.2 Evolution of SSA

In this article, we use two SSA definitions: specific surface area per unit volume $s$ and specific surface area per ice volume $SSA_V$. They are closely related through the ice volume fraction $\phi_i$:

$$SSA_V = \frac{s}{\phi_i} \tag{8}$$

We mainly work with the quantity $s$ for the rest of the article. However, we note that the quantity $SSA_V$ is more commonly used in the snow community (e.g., Matzl and Schneebeli, 2006), since it directly corresponds to the optical diameter.

The solution of the heat and mass diffusion equations (Eq. (1)-(3)) with boundary conditions (Eq. (4)-(7)) yields the spatially varying interface velocity $v_\mathbf{n}$ at any point on the ice-air interface $\Gamma$. As shown by (Drew, 1990; Krol and Löwe, 2018), this information, together with information about surface curvature, is sufficient to calculate the evolution of the SSA rigorously via surface averaging. As a result, for single grains or statistically homogeneous microstructures, the surface area evolution equation can be expressed as follows:

$$\dot{s} = 2\,s\,\overline{v_\mathbf{n}H} \tag{9}$$

Here the term $\overline{v_\mathbf{n}H}$, referred to as the growth rate in this article, is the product of the local interface velocity $v_\mathbf{n}$ and the local mean curvature $H$ averaged over the ice-air interface area (the surface average being indicated by an overline over the product).

Equation (9) is a linear homogeneous first-order ordinary differential equation and can be formally solved in closed form by separation of variables yielding

$$s(t) = s(0) \exp\left(2 \int_0^t \overline{v_{\mathbf{n}}H}(\tau)d\tau\right) \tag{10}$$

Equation (10) allows us to compute the SSA evolution from the growth rate $\overline{v_{\mathbf{n}}H}$ which must be computed from the solution of the 3D diffusion problem. This link between the SSA evolution and heat and mass diffusion equations is rigorous.

## 3   Numerical modeling

The end-goal of our numerical modeling is to simulate the SSA decrease of snow samples over time based on the pore-scale physics, and to compare this decrease to experimental observations. For that, we rely on time-resolved $\mu$CT images that were obtained under TGM conditions (Pinzer et al., 2012). These $\mu$CT scans provide (i) experimental data of the evolution of the SSA over time and (ii) snow-microstructures that can be used for our physical modeling. The computation of a vapor field using an FE simulation, combined with the local curvature of the snow sample, allows us to estimate $\overline{v_{\mathbf{n}}H}$ over a given snow microstructure. With Eq. 9, this yields the evolution of the SSA during a given time interval.

As we want to reproduce the SSA evolution of entire time series, our general workflow is as follows. For a given experimental time series, we initialize the first term $s^1$ of the simulated SSA values using the SSA deduced from the first $\mu$CT image of the experimental time series. Then, the second simulated SSA value $s^2$ is computed by applying the growth rate deduced from an FE simulation performed on the first $\mu$CT image. The procedure is then repeated to compute the $n^{\text{th}}$ term of the simulated SSA $s^n$ using the already known value $s^{n-1}$ and an FE simulation performed on the $n-1^{\text{th}}$ $\mu$CT image. The workflow and its different steps are detailed in the sections below and illustrated in Fig. 1.

### 3.1   $\mu$CT time lapse experiments

The numerical simulations were conducted on 4D image data of two TGM experiments (Series 1 and 2), which were previously acquired and already analyzed in (Pinzer et al., 2012) and (Krol and Löwe, 2016). In the experiments, a constant temperature gradient was applied by adjusting a snow sample's bottom and top temperature in an instrumented tomography sample holder, known as Snowbreeder (Pinzer and Schneebeli, 2009a). Series 1 lasted 384 h, while Series 2 lasted 665 h. The mean temperature $T$ of the sample and the temperature gradient $\nabla T$ are similar for both series: $T$ = -8.1 °C, $\nabla T = 47$ Km$^{-1}$ for Series 1 and $T = -7.6$ °C, $\nabla T = 55$ Km$^{-1}$ for Series 2. Both time series start from rounded grains with slightly different initial values of SSA and volumetric density, namely $SSA_V(t=0) = 20$ mm$^{-1}$, $\phi_i(t=0) = 0.31$ for Series 1 and $SSA_V(t=0) = 24$ mm$^{-1}$, $\phi_i(t=0) = 0.28$ for Series 2. For further experimental details, we refer to (Pinzer et al., 2012).

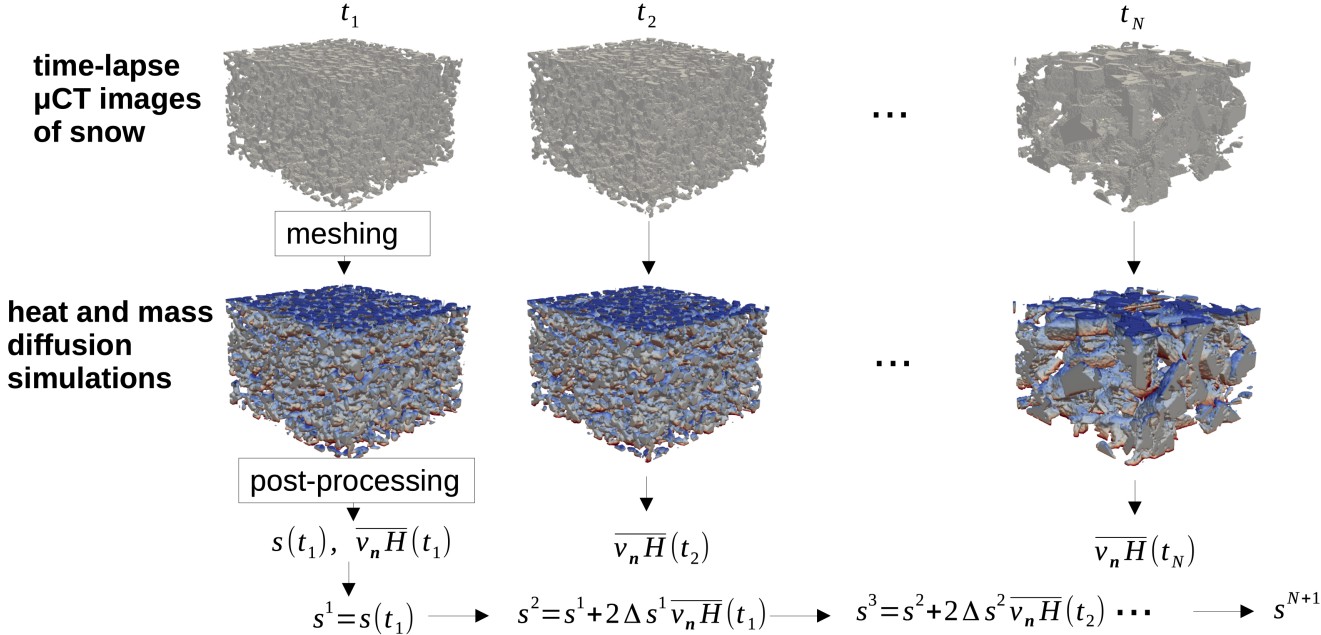

**Figure 1.** Schematic illustration of the workflow used in this study in order to compute the modeled SSA values $s^n$, $n = 2, \ldots, N+1$. For each 3D image of a $\mu$CT time-lapse series, a tetrahedral mesh is produced and a heat and mass diffusion simulation is conducted. The simulated interface growth velocity $v_\mathbf{n}$ is displayed in color in the Figure (blue corresponding to a receding interface and red to a growing interface). In the post-processing step, the growth rates $\overline{v_\mathbf{n}H}(t_n)$ are extracted and used to model the SSA evolution according to Eq. 18.

175  The $\mu$CT image data were taken from the snow sample every eight hours in time-lapse mode and segmented into binary images as described previously (Pinzer et al., 2012). These binary images are denoted by

$$I(t_n), \quad n = 1, 2, \ldots, 49 \qquad \text{for Series 1} \tag{11}$$

$$\tilde{I}(t_m), \quad m = 1, 2, \ldots, 84 \qquad \text{for Series 2} \tag{12}$$

at different time steps and are $300 \times 300 \times 196$ voxel images with voxel size length of $25 \cdot 10^{-6}$ m in Series 1 and of $18 \cdot 10^{-6}$

180  m in Series 2. This corresponds to samples of $7.5 \times 7.5 \times 4.9\,\text{mm}^3$ for Series 1 and $5.4 \times 5.4 \times 3.5\,\text{mm}^3$ for Series 2. Both series show the commonly observed decay of SSA (Taillandier et al., 2007; Pinzer and Schneebeli, 2009b; Calonne et al., 2014a).

### 3.2 FE solution of temperature and vapor fields

#### 3.2.1 Meshing

The production of an appropriate mesh that discretizes the air and ice domains, preserves the ice-air interface, and is fine

185  enough to get accurate numerical solution (without overloading computational resources) is a key requirement for our problem. To this end, we employ the open-source Computational Geometry Algorithms Library (CGAL) (The CGAL Project, 2022).

Specifically, we use the class `Polyhedral_mesh_domain_with_features_3` that implements a tetrahedral meshing of a domain bounded by polyhedral surfaces, which are preserved during the meshing process. The provided surfaces need to be closed and free of self-intersections. To obtain such surfaces, we extract the ice-air interface from the binary $\mu$CT data (Eq. (11) and (12)) following the procedure from (Krol and Löwe, 2018), namely by applying a Gaussian smoothing and the contour filter from the Visualization Toolkit (VTK) (Schroeder et al., 2006). However, by default, this procedure applied to $\mu$CT images yields a surface that is open at the boundaries of the domain. In order to obtain closed surfaces, we added a small air-padding (three voxel-thick) around the image. This allowed us to properly define a closed outer boundary suitable for meshing. As detailed below, we provided special care to ensure that the introduction of this artificial air-padding does not perturb the simulation within the snow microstructure itself. `MeshCriteria` parameters control the meshing algorithm in CGAL: Mesh tetrahedra are regulated by the radius-edge ratio upper bound of 1.5 and circumradius upper bound of 3 voxels, and triangles in the boundary surface mesh by the lower angular bound of $25°$ and radius upper bound of 0.75 voxels. These mesh parameters were manually fine-tuned through visual inspection. We have estimated the sensitivity of our results to the mesh parameters. We found that doubling the number of elements in the mesh impacted the simulated growth rate by about 10%. This is small in light of the dependence of the SSA values on the condensation coefficient $\alpha$ investigated in this study. Moreover, the very good agreement between an FE simulation and the analytical solution for a spherical problem (see Sect. 3.5) suggests that our meshing criteria yield an appropriate mesh. We save the mesh in four files listing the nodes, bulk elements, boundary elements, and header information, defining a mesh in the format of the FE software Elmer (Malinen and Råback, 2013). In addition, we computed the boundary weight on each mesh node $k$ of the triangulated ice-air interface $\Gamma_h$

$$\omega_k = \int_{\Gamma_h} \psi_k \, d\Gamma_h \tag{13}$$

where $\psi_k$ is the basis function assigned to the node $k$, so that the sum of all boundary weights $\omega_k$ gives the area of the whole boundary surface. Saving boundary weights is substantial for the computation of the interface velocities as surface integrals over the solution of heat and mass diffusion equations. For consistency and accuracy, employing the same integration scheme that underlies the FE solution is advantageous.

The FE meshes of this article are based on the whole available $\mu$CT images. We verified that these selected volumes were large enough to yield representative results. By varying sub-volumes extracted from the center of $\mu$CT images at the start and the end of both series ($I(t_1)$, $I(t_{49})$, $\tilde{I}(t_1)$ and $\tilde{I}(t_{83})$), we found that the simulated growth rate corresponds to a representative value for the sample sizes used in this study. This is consistent with the results of (Calonne et al., 2011) for thermal conductivity, that report representativeness for sample side-lengths between 2.5 and 5 mm.

### 3.2.2 FE solution

On the tetrahedral FE mesh with preserved surface, we solve heat and mass diffusion equations (Eq. (1) - (3)) employing open-source FE software Elmer (Malinen and Råback, 2013). For the simulation, we need to apply a given temperature gradient across the snow microstructure. However, due to the presence of artificial air-padding, directly applying the required tem-

perature gradient across the whole FE mesh (snow plus air-padding around the image) would result in a smaller temperature gradient within the snow itself (as the air is less conducting than the snow and thus concentrates the temperature gradient). In order to obtain the proper temperature gradient across the snow microstructure, the simulations are performed in two consecutive steps. First, the heat diffusion equation is solved over the whole FE mesh (snow plus air-padding around the image), and its result is used to estimate how a temperature gradient applied across the whole FE mesh translates into a temperature gradient within the snow microstructure itself. This allows us to determine a corrected temperature gradient to be applied across the whole FE mesh, in order to obtain the desired temperature gradient in the snow. Then, this corrected temperature gradient is used to solve the heat and mass diffusion equations with the appropriate temperature gradient across the snow microstructure. For the computation of heat and mass diffusion equations, we use the standard Elmer solvers `HeatSolver` and `AdvectionDiffusionSolver`, following Fourteau et al. (2021a). The equations are solved with the iterative biconjugate gradient stabilized method (BiCGSTAB; Van der Vorst, 1992) with an `ILU` preconditioner, meant to facilitate the numerical solving by performing an incomplete LU factorization (Saad, 2003). The maximum number of iterations is set to 2000, and the convergence tolerance to $10^{-10}$ for the heat diffusion equation and $10^{-12}$ for the mass diffusion equation. The correct temperature gradient across the domain is applied by setting top and bottom temperatures to

$$T_{\text{top}} = T - \frac{h \cdot \nabla T}{2}, \qquad T_{\text{bottom}} = T + \frac{h \cdot \nabla T}{2}, \tag{14}$$

where $T$ and $\nabla T$ are the experimental temperatures and temperature gradient and $h$ is the total height of the sample.

For the vapor boundary condition, we combine the Stefan condition (Eq. (4)) by neglecting the $\rho_v v_{\mathbf{n}}$ term due to $\rho_v \ll \rho_i$, and the Gibbs-Thomson equation (Eq. (7)) to obtain a Robin boundary condition at the ice-air interface

$$D_v \mathbf{n} \cdot \nabla \rho_v = \alpha v_{\text{kin}}[\rho_v - \rho_{v,s}(1 + d_0 H)], \qquad v_{\text{kin}} \approx 140 \, \text{m s}^{-1}, \, d_0 \approx 10^{-9} \, \text{m} \tag{15}$$

Here, the equilibrium water vapor concentration is given by the Clausius-Clapeyron relation, corrected for the Gibbs-Thomson effect: (Fourteau et al., 2021a)

$$\rho_{v,s} = \frac{M}{RT} P_0 \exp\left(\frac{L}{R}\left(\frac{1}{T_0} - \frac{1}{T}(1 + d_0 H)\right)\right), \qquad \frac{mP_0}{R} \approx 1.32 \, \text{kg K m}^{-3}, \, \frac{L}{R} \approx 6140 \, \text{K}, \, T_0 \approx 273 \, \text{K} \tag{16}$$

where $M$ is the molar mass of water, $R$ is the ideal gas constant, $L$ is the latent heat of sublimation of ice, $T_0$ is the reference temperature and $P_0$ is the saturation pressure at $T_0$. In contrast to (Calonne et al., 2014b; Fourteau et al., 2021a), the curvature term $d_0 H$ is not neglected. The mean curvature $H$ on the surface mesh is obtained following (Krol and Löwe, 2018) involving the shape operator computed with the normal vector field. We compute the field of normal vectors $\mathbf{n}$ using the dedicated routine of Elmer. It was found to be more reliable than VTK computations performed on the CGAL mesh, as the latter sometimes produces areas with reversed normal vectors.

Finally, the required local interface velocity $v_{\mathbf{n}}$ is computed using the vapor flux deduced from the FE simulation. For this, we use the `Calculate Loads` option of Elmer that provides the vapor flux $f_k$ (expressed in kg s$^{-1}$) at each node $k$ of the ice-air interface. Dividing by the associated boundary weight $\omega_k$ yields the corresponding deposition/sublimation flux (expressed in kg m$^{-2}$ s$^{-1}$) over the ice-air interface. Thus, the interface velocity at node $k$ is recovered from the simulation as

$$(v_{\mathbf{n}})_k = -\frac{f_k}{\omega_k \rho_i} \tag{17}$$

### 3.3 Post-processing and derived SSA evolution

For a given time sequence $t_1$, $t_2$, … $t_N = t$ with $t_N = N\Delta$ of available $\mu$CT images (Eqs. (11), (12)) and available FE solutions of the vapor field, the SSA is inferred from the discretized solution of Eq. (9) obtained with the forward Euler method

$$s^{n+1} = s^n + 2\Delta s^n \, \overline{v_{\mathbf{n}} H}(t_n)) \tag{18}$$

where $s^n := s(t_n)$. The rates $\overline{v_{\mathbf{n}} H}(t_n)$ are calculated for each time step $t_n$ as surface integrals from the 3D FE solution. For that, we use the VTK package and first cut off the small air padding on the sides using `vtkClipDataSet`. Then, the triangulated ice-air interface is extracted. The local interface velocity $v_{\mathbf{n}}$ is directly taken from the FE simulation using Eq. (17). For the local curvature $H$, we employ the image analysis derived in Krol and Löwe (2018), which is based on the shape operator, as explained in Section 3.2.2. Finally, the surface integration for the average in $\overline{v_{\mathbf{n}} H}(t_n)$ takes into account the variable element size of the triangular mesh of the ice-air interface.

### 3.4 Stochastic model for the discretization error

While the combination of the theoretical solution of the diffusion equation and the SSA evolution is, in principle, exact, the 4D image data processing and the derived SSA are subject to experimental and processing errors. These errors could be of various origins, for instance due to uncertainties related to the estimation of the ice-air interface from the $\mu$CT scans or to errors related to the numerical FE discretization. When simulating on the temporal evolution of SSA over time, these errors will accumulate and be propagated into the modeled decrease of $s(t)$. To analyze how these errors translate to the overall SSA decrease, and how this depends on the temporal resolution, we resort to a simple stochastic error treatment. To this end, we write the rigorous representation of the SSA evolution from above as

$$s^{\text{true}}(t) = s(0) \exp\left( 2 \int_0^t d\tau \, r^{\text{true}}(\tau) \right) \tag{19}$$

and indicate that the true decay rate $r^{\text{true}}(\tau) = \overline{v_{\mathbf{n}} H}$ is in general unknown and concealed by errors. In the simplest setting, one would expect that the predicted SSA can, therefore, be written as

$$s(t) = s(0) \exp\left( 2 \int_0^t d\tau \, r(\tau) \right) \tag{20}$$

where the measured rate $r(\tau)$ differs from the true rate by a noise term via

$$r(\tau) = r^{\text{true}}(\tau) + \delta r(\tau) \tag{21}$$

Here, $\delta r$ is an additive noise, representing uncorrelated errors (for now of unspecified origin), which affects the computations at each time step. This implies that, on average, the computed SSA estimates are not equal to the true value $s^{\text{true}}$ but rather to

$$s(t) = s^{\text{true}}(t) \left\langle \exp\left( 2 \int_0^t d\tau \, \delta r(\tau) \right) \right\rangle \tag{22}$$

where $\langle \bullet \rangle$ denotes the average with respect to the additive noise. For a finite time step $\Delta$, the discrete solution can now be written as

$$s_\Delta(t) = s^{\text{true}}(t) \left\langle \exp\left(2\Delta \sum_{i=1}^{N} \delta r(t_i)\right) \right\rangle \tag{23}$$

where the dependence on the time step $\Delta$ has been made explicit in the notation. For uncorrelated measurement errors, we assume $\delta r_i := \delta r(t_i)$ to be i.i.d. Gaussian random variables with zero mean and variance $\langle \delta r_i^2 \rangle = \sigma^2$. Since the averaged exponential in Eq. (23) is nothing but the characteristic function of $\delta r_i$, the average can be readily calculated and written as

$$s_\Delta(t) = s^{\text{true}}(t) \exp(2\Delta\sigma^2 t) \tag{24}$$

Since the truth in Eq. (24) is unknown, *absolute* errors are a priori not accessible. However, we can exploit Eq. (24) to define a *relative* error metric that quantifies the differences due to different temporal resolutions when integrating Eq. (19). To this end, we define

$$\varepsilon(\Delta, \Delta', t) := \frac{(s_\Delta(t) - s_{\Delta'}(t))^2}{s_\Delta(t)^2} \tag{25}$$

which allows us to assess the influence of using different time steps in the SSA evolution. By simplifying Eq. (25) we infer

$$\varepsilon(\Delta, \Delta', t) = [1 - \exp(2|\Delta - \Delta'|\sigma^2 t)]^2 \tag{26}$$

which relates simulated SSA differences at time $t$ to the temporal resolution of the model and the variance of the measurement error $\sigma$.

### 3.5 Workflow validation: Growth of a spherical shell

We set up a complex numerical workflow that starts from a voxel image, computes the interface velocity $v_{\mathbf{n}}$ from an FE simulation, and eventually yields the growth rate $\overline{v_{\mathbf{n}}H}$ after surface integration. In order to validate the entire workflow, we consider a test case that can be compared to an analytical solution. To this end, we employ the classical situation of the Laplace equation in a spherical shell for the vapor concentration $\rho_v(r)$ with radial coordinate $r$ around a spherical particle with radius $R$ with fixed vapor concentration $\rho_\infty$ applied at the outer shell at distance $R_\infty$. A Robin boundary condition (Eq. (15)) is applied at the inner surface of the sphere, under the form $D_v \mathbf{n} \cdot \nabla \rho_v = \alpha v_{\text{kin}}[\rho_v - \rho_{v,s}]$, with $\rho_{v,s}$ a constant value smaller than $\rho_\infty$. Note that this problem is temperature-independent and is fully determined by the radius of the shells, and the values of $\rho_\infty$ and $\rho_{v,s}$. In this case, the interface velocity is known analytically (e.g., (Carslaw and Jaeger, 1986)), and due to spherical symmetry, the growth rate averaged over the surface is given by the value of the solution at $r = R$, via

$$\overline{v_{\mathbf{n}}H} = \frac{v_{\mathbf{n}}}{R} = \frac{D_v}{\rho_i} \frac{\rho_\infty - \rho_{v,s}}{R\left(R - \frac{R^2}{R_\infty} + \frac{D_v}{\alpha v_{\text{kin}}}\right)} \tag{27}$$

This analytical solution is compared to the numerical solution as follows. We start from a voxel image representation of the spherical shell as illustrated by the inner sphere in Fig. 2a, where the inner radius is set to $R = 10$ voxel and the outer radius set

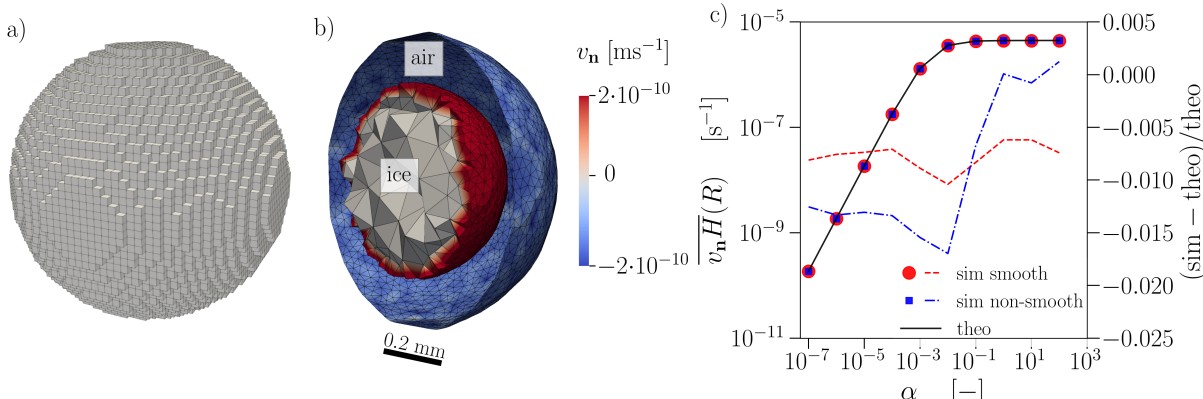

**Figure 2.** a) Voxeled sphere obtained from a binary image and used to constrain the problem. b) Clip of the outer and inner spherical shells with visible elements colored by the interface velocity $v_{\mathbf{n}}$ (sublimation in blue, deposition in red). c) Comparison of the growth rate $\overline{v_{\mathbf{n}}H}$ on the inner radius $R$ of theoretical (theo) and simulated (sim) solution of the spherical shell test case for different values of the condensation coefficient $\alpha$. Two different surface mesh qualities with (smooth) and without (non-smooth) smoothing are employed. The red dots, blue squares and black solid line correspond to $\overline{v_{\mathbf{n}}H}$ on the left y-axis while the dashed red and blue lines correspond to simulation error on the right y-axis.

to $R_{\infty} = 15$ voxel with a voxel size of $18\,\mu m$, corresponding to inner and outer radii of $0.18$ and $0.27\,mm$, respectively. In this way, the length scales of the test case are in a similar order of magnitude as the real microstructures considered later. Closed triangulated inner and outer sphere surfaces are created by applying the contour filter, which is subsequently passed as input to the CGAL volume meshing. A representation of the tetrahedral volume mesh obtained from CGAL and the corresponding triangular surface meshes are shown in Fig. 2b, where the volume mesh of the air space between the sphere has been left out for visual clarity. The slightly flattened regions on the sides of the sphere due to the original representation on a cubic lattice are still visible. The figure also reveals that the obtained CGAL mesh size is adaptive, i.e., in the vicinity of the interface, element sizes are reduced. After solving the vapor equation, with appropriate boundary conditions, we obtain the interface velocity $v_{\mathbf{n}}$, shown in Fig. 2b. As expected, we observe a positive velocity on the inner shell, corresponding to vapor deposition, and a negative velocity on the outer shell, corresponding to sublimation. We then use our standard post-processing procedure to calculate the averaged growth rate $\overline{v_{\mathbf{n}}H}$ as an integral over the triangulated surface of the inner sphere with local curvatures and interface velocities as described previously in Sect. 3.2.2 and 3.3. Since we shall later focus on variations as a function of the condensation coefficient, we have repeated this procedure for ten different values of $\alpha$. We also used two slightly different mesh quality parameters of the CGAL mesher to assess the sensitivity of the smoothness of the surface compared to the standard setup. The results of the validation are shown in Fig. 2c, yielding an excellent agreement of the numerical workflow with the analytical results for either smoothness. The results demonstrate that the choice of meshing and solver parameters leads to

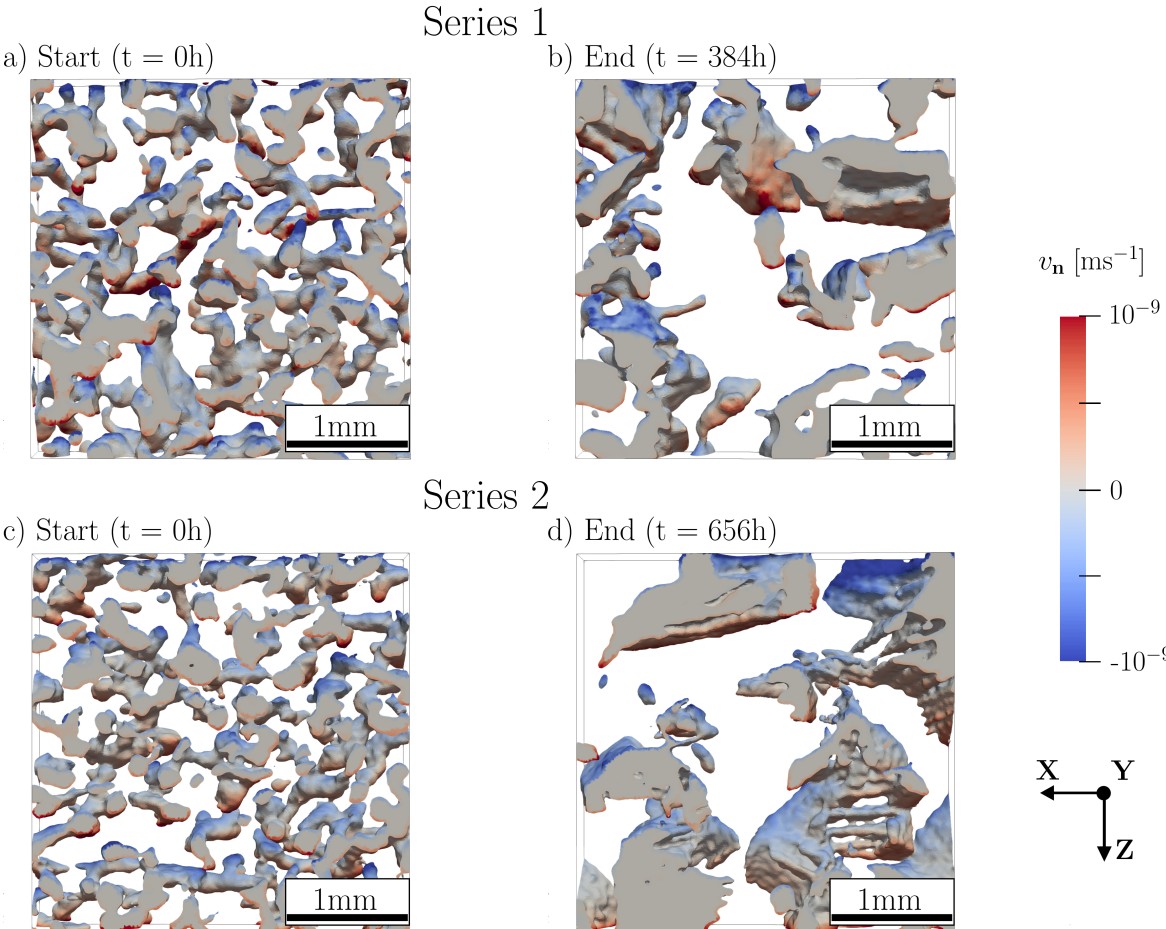

**Figure 3.** Evolution of the ice-air interface colored by interface velocity $v_\mathbf{n}$ demonstrated on cutouts of the length of 3.5 mm for a)/b) Series 1 and c)/d) Series 2.

reliable numerical results. The agreement provides confidence in the correctness of the implementation of the entire workflow, which is now applied to the 4D image data of TGM.

## 4    Results

### 4.1    Overview

As an overview and for a visual inspection of the microstructures and the rates derived from the FE solution, we show in Fig. 3
the initial and the final microstructure of both experimental series, each colored by interface velocity $v_\mathbf{n}$ (computed using Eq. (17)). This reveals the morphological differences at the end of both experiments, where the longer experiment (Series 2) has

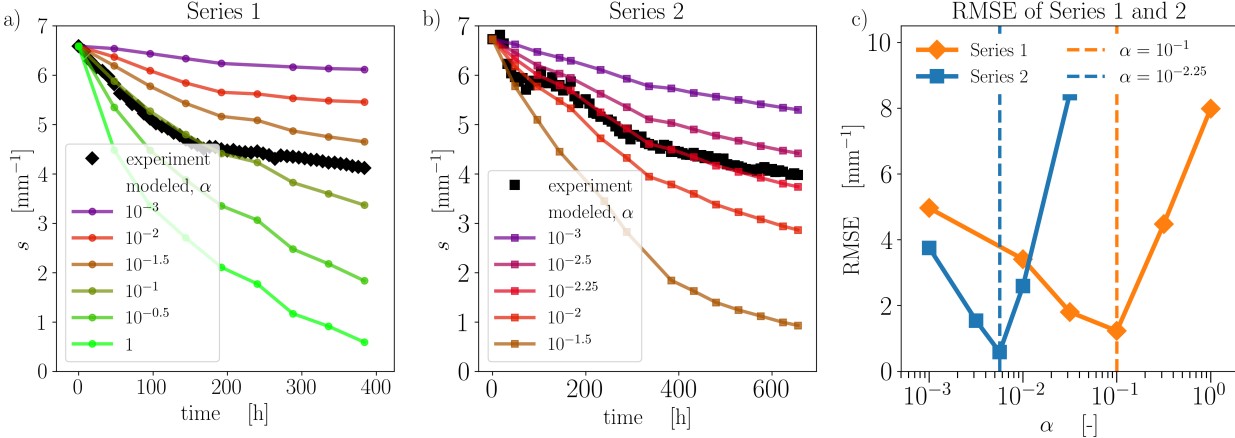

**Figure 4.** Time evolution of the SSA $s$ experimental and modeled with a varying condensation coefficient $\alpha$ for a) Series 1 and b) Series 2. c) RMSE for both series.

evolved into a more pronounced depth hoar state with enhanced formation of cup crystals (Pinzer et al., 2012). The simulations from Fig. 3 were carried out for the kinetic parameter $\alpha = 10^{-1}$ for Series 1 and $\alpha = 10^{-2.25}$ for Series 2 as showing the best root mean square error (RMSE) agreement in Fig 4c that is described in detail in the following section. As suggested by the

analytical solution (Fig. 2c), or the sensitivity of the vapor fluxes by (Fourteau et al., 2021a), the simulated SSA rates are highly sensitive to the condensation coefficient $\alpha$.

### 4.2   Coarse temporal resolution modeling: $\alpha$ estimation

In the first step, we compare the temporal evolution of the SSA $s$ between experimental data and the model using a large time step for the modeled data. For that, we downsample the experimental $\mu$CT time series to match the coarse temporal resolution

and only perform FE simulations on those. Specifically, the modeled SSA values are computed with a coarse time resolution of $\Delta = 48$ h for Series 1 (corresponding to 9 temporal points) and $\Delta \approx 60$ h for Series 2 (corresponding to 15 temporal points). This reduction in numerical effort allows us to perform a sensitivity study and estimate a value for the condensation coefficient $\alpha$ that best matches the experimental data. A fixed constant $\alpha$ is used for each simulation. The range of $\alpha$ varies from $10^{-3}$ to 1 for Series 1 and from $10^{-3}$ to $10^{-1}$ for Series 2. For the comparison with these simulated data, we simply use

all available experimental SSA data (acquired for a temporal resolution of 8h). The results are shown in Fig. 4a,b. We note that a few simulation points are missing in Fig. 4, due the non-convergence of the FE solver. That being said, these missing points do not modify the overall decay of the simulated SSA time series. The best visual agreement between the experimental and modeled data is found for $\alpha_{\mathrm{Series\,1}}^{\mathrm{best}} = 10^{-1}$ for Series 1 and $\alpha_{\mathrm{Series\,2}}^{\mathrm{best}} = 10^{-2.25}$ for Series 2. For Series 1, the initial stage of the modeled curve with $\alpha_{\mathrm{Series1}}^{\mathrm{best}}$ is close to the experimental data, while the final stage significantly underestimates the observed

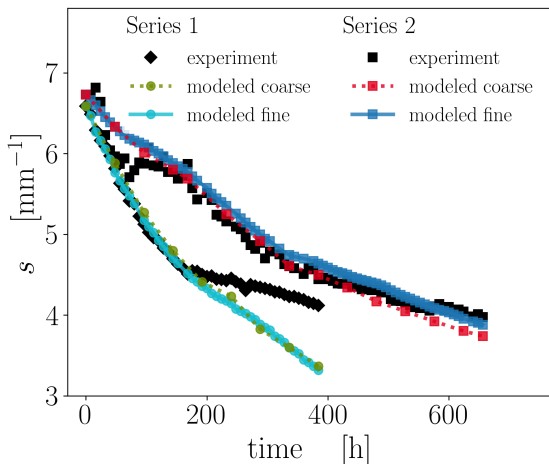

**Figure 5.** The time evolution of the SSA $s$ for both series with coarse and fine temporal resolution for the best previously found values $\alpha_{\text{Series 1}}^{\text{best}}, \alpha_{\text{Series 2}}^{\text{best}}$.

SSA. The same trend can be seen in Series 2, less prominent though. The experimental data of Series 2 reveals significantly more fluctuations in the initial phase, which is naturally not captured by the coarse resolution modeling.

To assess the accuracy of modeled data quantitatively, the RMSE is computed according to

$$\text{RMSE} = \sqrt{\frac{\sum_{n=1}^{N}(s_{\text{exp}}^{n} - s_{\text{mod}}^{n})^2}{N}} \tag{28}$$

where $N$ is the number of time steps involved in the modeling. The results are shown in Fig. 4c. The minimum of the RMSE
curve coincides with the best visual agreement, i.e., $\alpha_{\text{Series 1}}^{\text{best}} = 10^{-1}, \alpha_{\text{Series 2}}^{\text{best}} = 10^{-2.25}$. The difference between both optimal alpha values is one order of magnitude. Since the final stage of the modeled curve for Series 2 does not drop as much as for Series 1, the RMSE minimum for Series 2 is lower despite higher data scattering.

### 4.3 Impact of temporal resolution

To assess the impact of temporal resolution on the modeled decrease of SSA, we performed simulations with a time step refined
down to the time interval between two $\mu$CT images, namely 8 h. Based on results from the previous subsection, the simulations for the fine temporal resolution are carried out for the condensation coefficients $\alpha_{\text{Series 1}}^{\text{best}}, \alpha_{\text{Series 2}}^{\text{best}}$ that were obtained by RMSE optimization of the coarse resolution modeling. The results are given in Fig. 5. For Series 1, the fine resolution curve essentially coincides with the coarse one. The differences are slightly enhanced for Series 2, where the fine resolution curve lies slightly above the coarse one. The good agreement between the coarse and fine resolution simulations suggests that the coarse time
step used in the previous section is sufficient to estimate the optimal $\alpha$ values.

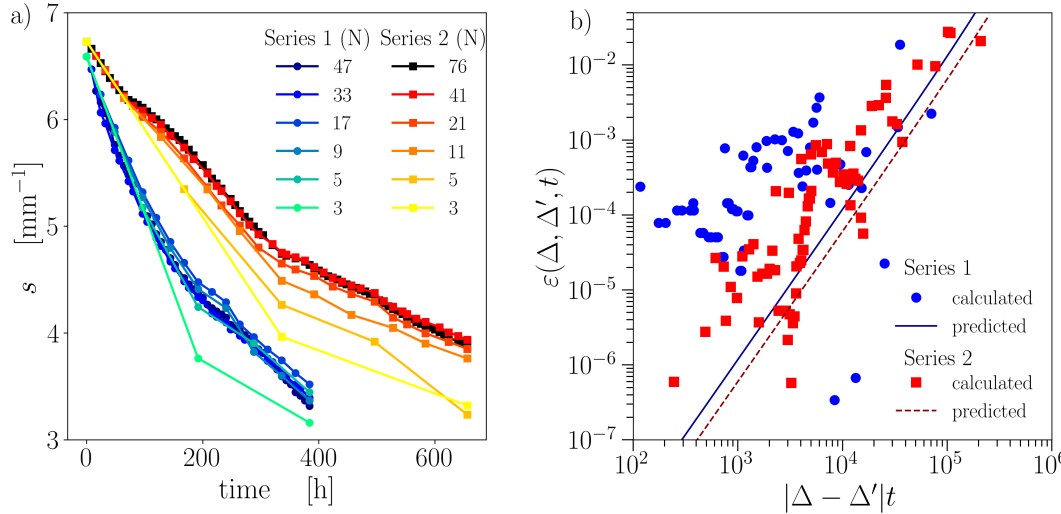

**Figure 6.** a) Different temporal resolutions of Series 1 and 2. b) The corresponding temporal resolution error $\varepsilon$ calculated via Eqs. (25) and (26).

These modeled SSA differences due to different temporal resolutions can now be further assessed through the error metric from Eq. (25). To this end, we fix the values of $\alpha$ to the optimal values found in the previous section and compute the SSA evolution for various temporal resolutions. We choose different numbers of time steps $N$ such that our model provides the time evolution of the SSA $s(t_n)$ with $n = 1, 2, \ldots N$ for different temporal resolutions $\Delta, \Delta'$ where $\Delta = t_N/(N-1)$ (see Fig. 6a). On the one hand, this allows us to calculate the error metric from Eq. (25) using the model results alone. The results are shown in Fig. 6b as solid markers for the two series. On the other hand, the error metric can also be independently estimated using the stochastic error model of Eq. (26) for the given variance $\sigma$. Fitting the variance using the least squares method on the modeled data leads to values $\sigma_{\text{fit}} = 0.0007$ and $0.0006$ for Series 1 and 2, respectively, and the results are shown in Fig. 6b as lines. These values are of the same order of magnitude as the variance computed as $\dot{s}/(2s)$ from the measurements: $\sigma_{\text{mes}}$ $= 0.0005$ and $0.0007$ for Series 1 and 2, respectively. Both estimations of the impact of the temporal resolution on the error metric are in reasonable agreement. Series 2 shows a significant difference in error between the coarsest and finest temporal resolutions, both from simulations (red markers) or according to the $\mu$CT data (red line). On the contrary, the simulation-based estimation of Series 1 (blue markers) does not drop as much for the finest temporal resolutions. This comes from the fact that the modeled SSA evolution using our finest and second-finest temporal resolution substantially differ. Overall, the error metric's usage indicates that the temporal resolution's impact on the SSA evolution remains relatively small, with errors below 1%.

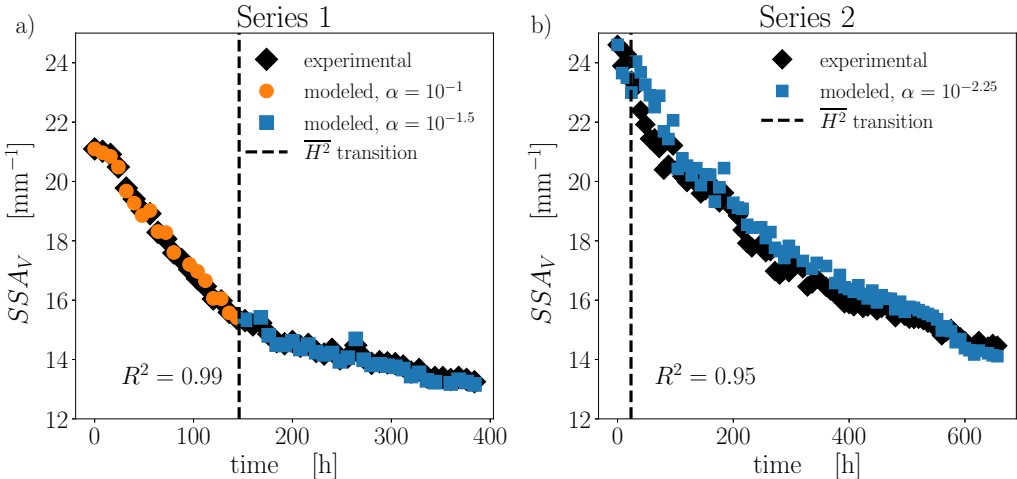

**Figure 7.** Comparison of experimental and modeled SSA time evolution. a) Series 1 with $\alpha_{\text{Series 1}}^{\text{best}}$ for $t \le 160$ h and $\alpha = 10^{-1.5}$ for $t > 160$ h. b) Series 2 with $\alpha_{\text{Series 2}}^{\text{best}}$.

### 4.4 Signatures of a transition in $\alpha$ during TGM

Since we obtain a good agreement between experimental and modeled data for Series 1 only in the initial stage, additional simulations were conducted to explore this further. As previously shown in (cf. Krol and Löwe, 2018, Fig. 6), the Series 1 undergoes a morphological transition at around $t \approx 160$ h, where up-facing and down-facing surfaces can be morphologically distinguished by their curvature distribution. From this time on, the second moment $\overline{H^2}$ of up-facing and down-facing surfaces split up to follow a different dynamics. Such a behavior during TGM is known from other work (Calonne et al., 2014a; Granger et al., 2021) and reflects the predominant emergence of facets on down-facing surfaces while the up-facing (sublimating) surfaces remain rounded. Here, we show that this morphological transition during TGM is consistent with a transition in the effective condensation coefficient $\alpha$ that governs the SSA decay. To reveal the different kinetic behavior of Series 1 in the initial and final stages, we set the transition to $I(t_n), n \ge 20$, i.e. t = 160 h, and performed independent optimization of the condensation coefficient. Very good agreement with the coefficient of determination $R^2 = 0.99$ is achieved when the condensation coefficient is set to $\alpha = 10^{-1.5}$ for the final stage. The results for the optimal parameters are shown in Fig. 7a. While the transition is also present in Series 2 (Fig. 6 Krol and Löwe, 2018), it occurs already very early in the time series after $t \approx 24$ h, cf. Fig. 7b. This is consistent with the observation that only one value of $\alpha$ is sufficient to match the measured data for Series 2. Since the initial stage in Series 2 is subject to higher fluctuations, an independent optimization of another $\alpha$ after a few time steps is inconclusive. Overall, this leads to the slightly reduced coefficient of determination $R^2 = 0.95$ for Series 2. Fig. 7 summarizes the best possible match we obtained for the SSA in the highest resolution within the developed method.

## 5   Discussion

### 5.1   Modelling the SSA evolution from first principles

We have set up a numerical model that can simulate the evolution of one of snow's most fundamental microstructural parameters, the SSA, from 3D $\mu$CT images. The model is based on the established theoretical description of snow metamorphism through coupled heat and mass diffusion at the pore scale (Kaempfer and Plapp, 2009; Calonne et al., 2014b). The solution of the diffusion problem thereby extends previous work characterizing TGM from $\mu$CT images (Flin and Brzoska, 2008; Pinzer et al., 2012; Krol and Löwe, 2016), where vapor fluxes were estimated only within the assumption of local equilibrium at the interface. Under this assumption, fluxes can be estimated from temperature fields and curvatures alone without explicitly solving the vapor equation. Our diffusion model is essentially physically equivalent to (Kaempfer and Plapp, 2009) in the steady-state limit and has been used previously (Fourteau et al., 2021a).

The actual novelty of our work is the combination of the numerical solution of the heat and mass diffusion with the exact surface area evolution equation (Krol and Löwe, 2018). This combination allows us to rigorously validate the SSA dynamics without explicitly evolving the ice-air interface in 3D space. This approach is thus complementary to 4D microstructure evolution models such as (Kaempfer and Plapp, 2009) or (Bouvet et al., 2022). The advantage of including the surface area equation (Eq. (9)) into the analysis is the possibility of isolating the relevant growth rate $\overline{v_{\mathbf{n}}H}$, either for constructing a stochastic error analysis (Sec. 3.4) or validation with analytical results (Sec. 3.5).

The model still requires considerable numerical resources, including volume meshing of the microstructure, the FE solution of heat and mass diffusion equations taking into account kinetic effects of crystal growth, the extraction of the interface velocity $v_{\mathbf{n}}$ from the vapor field and the subsequent integration of the surface area equation. Nevertheless, we were able to reproduce the decay of the SSA during TGM for the first time from "first principles", i.e. using a physical model and the actual microstructure without adjusting free parameters (in contrast to (Legagneux et al., 2004; Domine et al., 2007; Taillandier et al., 2007)). The only unknown (physical) parameter in the model is the condensation coefficient, which characterizes vapor deposition and sublimation kinetics.

### 5.2   The condensation coefficient $\alpha$

We have demonstrated that the SSA evolution in the model is highly sensitive to the condensation coefficient $\alpha$ (see Fig. 4). The best agreement (see Fig. 7) is obtained for values of $10^{-3} < \alpha < 10^{-1}$ (slightly different for the two time series) that fall in the intermediate range (Fourteau et al., 2021a) of possible values. This intermediate range of kinetics is neither compatible with the assumption of slow kinetics underlying the homogenization from (Calonne et al., 2014b) nor the assumption of infinitely fast kinetics, which was previously used to compute $v_{\mathbf{n}}$ from local temperature gradients (Krol and Löwe, 2016). While infinitely fast kinetics was already suggested to be inconsistent with the present experimental data sets (Krol and Löwe, 2018), this is now confirmed here from the estimated range for the values of $\alpha$. From these results, we conclude that precise information about $\alpha$ is essential and modeling the SSA during TGM solely using geometry and temperatures/gradients and neglecting kinetic effects (Flanner and Zender, 2006) cannot be justified.

It is well known that $\alpha$ is difficult to measure experimentally. This is explained in (Libbrecht, 2005) and can be easily understood from Fig. 2c: When $\alpha$ is commonly measured through the inversion of interface velocity $v_{\mathbf{n}}$ data, the saturation form of the curve for the growth rate $\overline{v_{\mathbf{n}}H}$ as a function of $\alpha$ implies significant uncertainties on $\alpha$ even for minor errors in the growth rate in the saturation region, where diffusion dominates. Our methodology can be considered as a new (but similar) possibility of retrieving $\alpha$ by comparing simulated SSA evolution curves with experimental ones. From the reasoning given above, a high uncertainty should be expected. Surprisingly, the optimization (Fig. 4) reveals a rather sharp minimum. A similar procedure for obtaining $\alpha$ from the comparison of measured and modeled SSA curves was recently suggested by (Bouvet et al., 2022), where a value of $\alpha \approx 9.8 \times 10^{-4}$ was obtained from a comparison of a phase field model with experimental data in isothermal metamorphism. The latter work put forward an interesting alternative route to the optimization of $\alpha$ from experimental data by means of dimensional analysis. So, instead of conducting many simulations of different $\alpha$ (as done here), the same results could be obtained through non-dimensionalization and a single simulation. However, the temperature gradient case considered here is governed by two different time scales instead of only one in the isothermal case (Bouvet et al., 2022), which renders this approach less straightforward in our case. When comparing our results to other data, we see that the obtained values $10^{-1}, 10^{-1.5}$ for Series 1 and $10^{-2.25}$ for Series 2 lie in the commonly found range of $10^{-3} < \alpha < 10^{-1}$ (Libbrecht and Rickerby, 2013) which is also used by (Kaempfer and Plapp, 2009). They are slightly higher but in a similar order of magnitude as reported in (Fourteau et al., 2021a; Bouvet et al., 2022). In contrast, the condensation coefficient from Jafari et al. (2020) translates to $\alpha \approx 5 \cdot 10^{-7}$, which is significantly below this range.

In addition to the fact that both experimental series are apparently governed by a different condensation coefficient (Fig. 4), we have provided evidence (Fig. 7) that the condensation coefficient may even change during a single experiment. To comprehend this finding, we recall that in snow, different parts of the ice-air interface belong to different crystallographic orientations and habits (rounded vs. faceted). Both have different attachment mechanisms and, therefore, different $\alpha$ (Libbrecht, 2005). Using a single, constant value of $\alpha$ that does not vary over the surface (as done here) must be therefore understood as an *effective* kinetic coefficient. This effective coefficient can capture actual micro-scale variations of $\alpha$ since a very good agreement for the SSA (as an integral property) is still obtained. It is quite remarkable that despite large variations of the condensation coefficient at the micro-scale, their collective behavior can be appropriately described through the use of a single $\alpha$ value. Indeed, in principle, the assumption of a constant $\alpha$ in Eq. 15 must be questioned on physical grounds. On facets, one expects that $\alpha$ is significantly reduced by orders of magnitude with a non-linear dependence on the ambient vapor field/supersaturation (Saito, 1996). Since facets cover only a fraction of the surface, this may explain why only a moderate drop in the effective $\alpha$ (Fig. 7) is observed instead. Further substantiation of this hypothesis in future work is feasible even without crystal orientation measurements such as (Granger et al., 2021). The surface area evolution equation (Eq. (9)) and the pore scale diffusion model can be easily extended to deal with spatially varying condensation coefficients on the ice-air interface and corresponding surface area sub-classes (e.g., up-facing and down-facing). Such a setup would allow us to validate the hypothesis for the condensation coefficient transition here. Then it would be beneficial to include higher order interfacial properties like $\overline{H}, \overline{H^2}$ explicitly in the validation. This is, however, at the cost of evaluating higher order rate terms.

## 5.3 Propagation of measurement errors

Our analysis has shown why high-quality $\mu$CT data is crucial for our methodology. The complex numerical workflow contains several sources of errors that may affect the predicted SSA evolution. First, experimental input data have a limited spatial and temporal resolution, which leads to missing structural and interface correlations between two consecutive images. With a different experimental setup, such as in (Calonne et al., 2015) a higher spatial resolution may be achieved, though. Second, the volumes of interest considered here could be larger, in particular for Series 2. This size might lead to some non-representativeness issues and small fluctuations in the measured SSA. This could explain the slightly noisy nature of the experimental parameter curves in Series 2 compared to Series 1. Third, all involved image analysis and simulation procedures come with additional numerical errors. While some uncertainties can be well controlled and assessed by testing the numerical workflow against analytical solutions (see Fig. 2), the existence of remaining errors is evident.

To address these errors and their impact on SSA modeling, we have exploited that the explicit SSA representation allows us to construct a stochastic error model (Sect. 3). This model predicts how the combination of temporal resolution $\Delta$, observation time $t$, and methodological errors (subsumed in the variance $\sigma$ of the $\mu$CT comparison data) affect the SSA prediction. The stochastic model is reasonably consistent with the observed convergence of the predictions under reduction of the time step (Fig. 6). The fact that errors can be quantitatively addressed even without knowing the true SSA is facilitated by the representation of the SSA as a differential equation (Eq. (9)). In the future, more sophisticated stochastic models should be envisaged and constructed from Eq. 9, which will further help to distinguish methodological noise and physics in the derived SSA dynamics.

## 5.4 Limitations and perspectives

Regarding model limitations besides the effective treatment of the condensation coefficient approach outlined above, we have neglected the latent heat term in the interface condition for the temperature equation (Eq. 6). This leads to a slightly simpler numerical situation where heat and vapor are coupled only one way, and the heat diffusion equation can be solved in advance. This strategy reduces the numerical cost of the method and and facilitates the convergence of the iterative solver used in the FE software. Despite this simplification, we still observe that the vapor solver had issues to converge for a few microstructures, which explains a few missing points in the modeled time series (e.g., Fig. 5). The convergence of the FE simulations depends on the employed mesh and on the value of $\alpha$. It could be facilitated by improving the mesh quality or increasing the maximum number of iterations. While this one-way coupling assumption eases the numeric, it was previously shown (Fourteau et al., 2021b) that for low density or fast kinetics, latent heat significantly contributes to the heat fluxes in snow and may thus likewise impact the volume averaged rate term $\overline{v_{\mathbf{n}} H}$. This should be carefully investigated for low-density $\mu$CT time series under TGM in the future, where the numerical solution will become more demanding. In general, it would be advantageous to extend the analysis to other data sets. Here, we have used only two TGM time series which have been well studied before (Kaempfer et al., 2005; Pinzer et al., 2012; Krol and Löwe, 2018). Evaluation of high-resolution TGM experiments with systematic variations of the control parameters (microstructure, temperature, and temperature gradients) would be desirable. This would allow us to

parameterize the relevant rate term $\overline{v_{\mathbf{n}}H}$ from the control parameters, which is the most promising way to proceed towards a physically based SSA equation in snow cover models.

## 6  Conclusions

We have addressed the SSA evolution in TGM within a rigorous framework that combines the surface area equation with pore-scale heat and mass diffusion simulations. The comparison to experimental $\mu$CT data allowed us to estimate effective condensation coefficients that led to good agreement of the simulations with the measurements without further adjustable parameters. This shows that the evolution of SSA can be understood from the first principles of pore-scale physics (diffusive heat and mass transports), provided that the effective condensation coefficient $\alpha$ is well-constrained. While this is a considerable step in understanding TGM our results highlight the importance of independent estimates of the condensation coefficient in snow, which is indispensable to proceed towards physically based SSA parameterizations in snow cover models.

*Code and data availability.*  We published simulation parameters and outputs for this study on the data portal EnviDat, doi:10.16904/envidat.492

*Author contributions.*  A.B. and H.L designed the study. A.B. and K.F. wrote the code. A.B. performed numerical computations. A.B, K.F. and H.L. discussed the data and wrote the manuscript. H.L. received the funding and supervised the study.

*Competing interests.*  The authors declare having no competing interests.

*Acknowledgements.*  A.B. and H.L. would like to thank Prof. Dr. Michael Lehning for fruitful discussions. The project was funded by the Swiss National Science Foundation (SNSF) under grant no. 200020_178831. KF current position is funded by the European Research Council (ERC) under the European Union's Horizon 2020 research and innovation program (IVORI, grant no. 949516). We are thankful to Z.R. Courville and T. Kaempfer for reviewing the manuscript and to M. Niwano for editing it.

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
