# Peer review of "A rigorous approach to the specific surface area evolution in snow during temperature gradient metamorphism"

_EGUsphere, 2023_

## Referee Comment (RC2)

[referee-annotated manuscript omitted]

---

## Author Comment (AC1)

**Response to RC2 from Thomas Kaempfer on egusphere-2023-1947**

We are thankful to Thomas Kaempfer for thorough and constructive review of our manuscript.

We have copied the comments from annotated PDF of the review, with the corresponding text line numbers, below in blue. Our responses are available in black below the comments. Proposed modifications to the manuscript are given in yellow highlighted italics. The bold text line numbers correspond to the original manuscript.

Best regards

Anna Braun on behalf of all co-authors

The paper presents a novel approach to model the evolution of the specific surface area (SSA) during temperature gradient metamorphism (TGM). It uses X-ray micro-computed tomography (mu-CT) images of snow in combination with a numerical solution of steady-state energy and mass conservation equations at the micro-structural scale and a surface area equation based on physics first principles. The only "free" parameter in the model is the vapor attachment coefficient alpha and it is proposed that SSA evolution can be predicted using an adequately chosen "effective" alpha.

The paper is generally well written with a strong emphasis on the numerical solution and analysis of error propagation. The strength and limitations of the approach are clearly presented.

Clarity could be improved by using more concise language and unified terminology (see minor comments in annotated PDF-file). The paper could further benefit by considering the following specific remarks. I suggest the paper to be revised accordingly before accepting it for publication.

**Specific remarks**

1. Introduction, line 36ff: While it is OK to quickly concentrate on the relevant mechanisms for the TGM situation studied in this paper (energy and mass conservation, attachment kinetics), I suggest justifying why other processes are of second order instead of "boldly" postulating that it is all about alpha.

It is indeed true that while heat and vapor diffusion together with vapor attachment kinetics are usually the processes used to study snow metamorphism from the pore scale, other processes (such a mechanical deformation or air advection) might interact metamorphism and the temporal evolution of the SSA. This will be clarified in the introduction by adding text **Line 36:**:

*"Physical models of snow metamorphism must comply with the ice crystal growth dynamics at the pore scale (Krol and Löwe, 2016), which includes heat and vapor diffusion, accommodated by attachment kinetics controlling the deposition and sublimation of water molecules onto the ice lattice (Colbeck, 1983; Libbrecht, 2005). Secondary effects on the temporal SSA evolution might be expected from other processes like mechanical deformation (Wang and Backer, 2013; Schleef et al., 2014), advection of air in the porosity (Ebner et al., 2016, Jafari et al., 2022). In this picture, one key parameter driving snow metamorphism is […]".*

We also propose to change the first paragraph of Sect. 2.1, **Lines 87-94** and explain that our motivation to neglect the potential influence of mechanics and air movement is twofold: (i) it corresponds to the basic mechanisms used to explain metamorphism in the most of the literature and is thus a good candidate in terms of minimum-required complexity, and (ii) it is consistent with the set-up under which the experimental data were acquired:

*"For an arbitrary snow structure, morphological changes during metamorphism are predominantly driven by the coupled diffusion of heat and mass together with ice-air interface evolution due to deposition and sublimation of vapor. In the following, we closely follow the descriptions by Kaempfer et al., (2009), Calonne et al., (2014), Krol and Loewe (2016), and Fourteau et al. (2020). We consider a representative snow volume at the micro-scale consisting of ice and air and denote the sub-domains occupied by the ice and air phase by $\Omega_i$ and $\Omega_a$, respectively. In the following, subscripts i and a denote quantities which are defined in the respective domains $\Omega_i$ and $\Omega_a$. Due to the separation of time scales between heat and mass diffusion in the pores and the evolution of the interface due to crystal growth, we employ the common assumption of small particle Péclet number (Libbrecht, 2005) and consider stationary heat and mass diffusion equations. Furthermore, we neglect the influence of mechanical deformation, as usually done in pore-scale metamorphism models (e.g., (Calonne et al., 2014b; Krol and Löwe, 2016)). We also neglect the potential presence of convection and air advection in the pore space. These assumptions are consistent with the experimental data used here, obtained under controlled laboratory conditions (Pinzer et al., 2012). They are also good candidates in terms of minimum-required complexity to model SSA evolution from pore-scale physics. The partial density of water vapor [...]".*

2.Presentation of the approach (e.g., Introduction, lines 70ff; or beginning of section 3, lines 159ff – possibly add a new sub-subsection at the beginning of sub-section 3.2): The coupling of the mu-CT images and the numerical models (Finite Elements and surface equation) should somewhere carefully be explained. If I understood it correctly, you have resp. do:

- time-laps mu-CT images of snow (4D)
- use a (sub-)set of 3D images as input to your numerical model
- for each chosen 3D image, pre-process and discretize (e.g., image processing, triangulation)
- determine some parameters from the geometry directly (e.g., surface curvature)
- determine other parameters from numerical modeling (e.g., growth velocity)
- compute SSA evolution using above; fit alpha to the experimental results

Yes, your description of the workflow is essentially correct. Specifically we:

1- Select a µCT time series with a given time resolution

2- Initialize the first term $s^1$ of the simulated SSA time series with the value deduced from the first µCT scan.

3- For each timestep $t_n$ of the time series, we compute the SSA growth rate associated with the corresponding microstructure (using a FE simulations).

4- Use the growth rate to prolong the SSA time series from $s^n$ to $s^{n+1}$.

A schematic summarizing this workflow is shown below.

[Figure]

- •Unclear to me are in particular:

•how many 3D images do you use for the modeling? When and how do you decide to "update" the micro-structure in your models (e.g., using a new image from the 4D series)? Or do you never do this (and always use the 1st image only)? In general, the discretization in time of the numerical model is unclear to me.

In the modeling, we use one 3D image for each timestep of a time series. As detailed above, for each timestep, we compute the evolution from $s^n$ to $s^{n+1}$ using the 3D image corresponding to timestep $t_n$. The first term $s^1$ of the SSA time series is initialized using the first 3D image, and the subsequent terms are computed using the growth rates deduced from the FE simulations on the 3D image of timestep $t_n$.

This will be specified in the text at the start of the Numerical Modeling Section **Line 140**, alongside a brief presentation of our simulation workflow:

*"The end-goal of our numerical modeling is to simulate the SSA decrease of snow samples over time based on the pore-scale physics, and to compare this decrease to experimental observations. For that, we rely on time-resolved µCT images that were obtained under temperature gradient metamorphism conditions (Pinzer et al., 2012). These µCT scans provide (i) experimental data of the evolution of the SSA over time and (ii) snow-microstructures that can be used for our physical modeling. The computation of a vapor field using a FE simulation, combined with the local curvature of the snow sample, allows us to estimate <$v_nH$> over a given snow microstructure. With Eq. 9, this yields the evolution of the SSA during a given time interval.*

*As we want to reproduce the SSA evolution of entire time series, our general workflow is as follows. For a given experimental time series, we initialize the first term $s^1$ of the simulated SSA values using the SSA deduced from the first µCT image of the experimental time series. Then, the second simulated SSA value $s^2$ is computed by applying the growth rate deduced from a FE simulation performed on the first µCT image. The procedure is then repeated to compute the $n^{th}$ term of the simulated SSA $s^n$ using the already known value $s^{n-1}$ and a FE simulation performed on the n-1$^{th}$ µCT image. This workflow and its different steps are detailed in the Sections below."*

We have also revised the start of Section 4.2 **Line 291** to better explain how the coarse temporal modeling is achieved:

*"In the first step, we compare the temporal evolution of the SSA s between experimental data and the model using a large time step for the modeled data. For that, we downsample the experimental µCT time series to match the coarse temporal resolution and only perform FE simulations on those. Specifically, the modeled SSA values are computed with a coarse time resolution of $\Delta$ = 48h for Series 1 (corresponding to 9 temporal points) and $\Delta$ ~60h for Series 2 (corresponding to 15 temporal points). This reduction in numerical effort allows us to perform a sensitivity study and estimate a value for the effective condensation coefficient α that best matches the experimental data. A fixed constant α is used for each simulation. The range of alpha varies from $10^{-3}$ to 1 for Series 1 and from $10^{-3}$ to $10^{-1}$ for Series 2. For the comparison with these simulated data, we simply use all available experimental SSA data (acquired for a temporal resolution of 8h). The results are shown in Fig.3a,b."*

•selection of appropriate (sub)volumes from the 3D images, pre-processing and volumetric averaging: how exactly are the volumes that feed into the numerical model selected? For several parameters, volumetric averaging is performed (e.g., SSA, curvature, alpha). Is the averaging volume always the same (e.g., the entire domain)? Are we sure to have a size large enough to be representative? For the kinetic coefficient, it is very late in the paper that the concept of "effective coefficient" is introduced. Maybe the concept(s) could be introduced and justified early in an overall context.

For all simulations, we have used the largest available volume. We have performed additional simulations to determine the representativeness of the growth rate computed with different sample sizes. It is shown in the graph below. The volumes used in the article correspond to the largest ones for each sample.

[Figure]

This suggests that the sample sizes used in this study are sufficient to yield representative results in terms of simulated growth rate. We will add this information **Line 179**:

*"The FE meshes of this article are based on the whole available µCT images. We verified that these selected volumes were large enough to yield representative results.. By varying sub-volumes extracted from the center of µCT images at the start and the end of both series ($I(t_1)$, $I(t_{49})$, $\hat{I}(t_1)$ and $\hat{I}(t_{83})$), we found that the simulated growth rate corresponds to a representative value for*

*the sample sizes used in this study. This is consistent with the results of Calonne et al. (2011) for thermal conductivity, that report representativeness for sample side-lengths between 2.5 and 5mm."*

Only the growth rate $\langle v_nH \rangle$ requires averaging. This averaging is performed on the whole triangulated ice-air interface of the sample. The different macroscopic quantities (SSA, growth rate, etc) are computed within the same snow volume to ensure consistency.

No averaging is performed on the condensation coefficient $\alpha$, as it assumed spatially-constant in the simulations. We will specify early in the text **Line 138** that the use of a spatially-constant alpha is akin to choosing an effective value.

We will specify the use of surface average in **Lines 72** and **127** and we will clarify the definition of the growth rate in **Line 130**:

*"Here the term $\overline{v_nH}$, referred to as the growth rate in this article, is the product of the local interface velocity $v_n$ and the local mean curvature H averaged over the ice-air interface area (the surface average being indicated by an overline over the product)."*

- reason for "numerical (?)" tricks like the "air padding" and iterative (2-times) solution of the energy conservation equation

Adding an artificial air-padding layer around the snow image is a numerical "trick" to obtain a surface mesh with closed outer boundaries using VTK, which is then required for meshing by CGAL. We will clarify the text about the artificial air-padding in **Line 160**:

*"To this end, we employ the open-source Computational Geometry Algorithms Library (CGAL) (The CGAL Project, 2022). Specifically, we use the class Polyhedral_mesh_domain_with_features_3 that implements a tetrahedral mesh of a domain bounded by polyhedral surfaces that are preserved. The provided surfaces need to be closed and free of self-intersections. To obtain such surfaces, we extract the ice-air interface from the binary µCT data (Eq. (11) and (12)) following the procedure from (Krol and Löwe, 2018), namely by applying a Gaussian smoothing and the contour filter from the Visualization Toolkit (VTK) (Schroeder et al., 2006). However, by default this procedure applied to µCT images yields a surface that is open at the boundaries of the domain. In order to obtain closed surfaces, we added a small air-padding (three voxel-thick) around the image. This allowed us to properly define a closed outer boundary suitable for meshing. As detailed below, we provided special care to ensure that the introduction of this artificial air-padding does not perturb the simulation within the snow microstructure itself."*

Solving heat equation twice is made to ensure that the presence of the air-padding does not interfere with the temperature gradient in the snow part of the mesh. We will clarify **Line 181** how we deal with air-padding in the simulation:

*"For the simulation, we need to apply a given temperature gradient across the snow microstructure. However, due to the presence of artificial air-padding, directly applying the required temperature gradient across the whole FE mesh (snow plus air-padding around the image) would result in a smaller temperature gradient within the snow itself (as the air is less conducting than the snow and thus concentrates the temperature gradient). In order to obtain the proper temperature*

*gradient across the snow microstructure, the simulations are performed in two consecutive steps. First, the heat equation is solved over the whole FE mesh (snow plus air-padding around the image), and its result is used to estimate how a temperature gradient applied across the whole FE mesh translates into a temperature gradient within the snow microstructure itself. This allows us to compute a corrected temperature gradient to be applied across the whole FE mesh, in order to obtain the desired temperature gradient in the snow. Then, this corrected temperature gradient is used to solve the heat and mass diffusion equations with the appropriate temperature gradient across the snow microstructure. For the computation of heat and mass diffusion equations, we use the standard Elmer solvers HeatSolver and AdvectionDiffusionSolver, following Fourteau et al. (2021a)."*

3. Error analysis (sections 3.4 / 4.3 / 5.3): While I do like this systematic error analysis, I find the focus very much on "time" related errors; however, I think that other errors (e.g., discretization in space) are at least equally important. These are captured and a bit discussed in section 5.3. Clearer statements – already in earlier sections – would help to better understand the strengths and weaknesses of the approach chosen.

The goal of Section of 3.4 is to propose a framework to analyze how stochastic errors in the growth rate estimation (whatever their origin) propagates and accumulates over time in the SSA. As mentioned later in the manuscript, it is meant to estimate how the temporal resolution and methodological errors combines into a given error for the SSA prediction at a give time horizon. We agree that other types of errors (such as a potential bias due to spatial discretization) are also important.

Regarding the quantification of errors, we have quantified the sensitivity of our results to the mesh resolution. This is shown in the graph below, and will be mentioned in the revised manuscript **Line 170**:

*"We have estimated the sensitivity of our results to the FE mesh. We found that doubling the number of elements in the mesh impacted the growth rate by about 10%. This is small in light of the dependence of the SSA values on the condensation coefficient α investigated in this study. Moreover, the very good agreement between a FE simulation and the analytical solution for a spherical problem (see Sect. 3.6) suggests that our meshing criteria yield an appropriate mesh."*

[Figure]

We also propose to more clearly state at the beginning of Section 3.4 that the stochastic model is meant to quantify how methodological errors accumulate over time in the SSA decrease, and the impact of the temporal resolution in this process. **Line 223**:

*"While the combination of the theoretical solution of the diffusion equation and the SSA evolution is, in principle, exact, the 4D image data processing and the derived SSA are subject to experimental and processing errors. These errors could be of various origins, for instance due to uncertainties related to the estimation of the ice-air interface from the μCT scans or to errors related to the numerical FE discretization. When focusing on the temporal evolution of SSA over time, these errors will accumulate and be propagated into the modeled decrease of s(t). To analyze how these errors translates to the overall SSA decrease, and how this depends on the temporal resolution, we resort to a simple stochastic error treatment."*

Line 7: suggestion: quantify the impact

We will change **Line 7** accordingly:

*"[…], we quantify the impact of these […]"*

Line 16: , normalized per volume. Suggestion: remove the last part of the sentence since you give many examples in the next one.

We will change **Line 16** accordingly:

*"The specific surface area (SSA) of snow is the interface area between ice and air in the microstructure of porous snow, normalized per volume. The SSA is a crucial parameter for [...]"*

Line 20: one key

We will change **Line 20** accordingly:

*"The SSA evolution in time is one key to […]"*

Line 30: well characterized uncertainties

We will change **Line 30** accordingly:

*"[…] computed within well characterized uncertainties due to […]"*

Line 37: maybe you could shortly explain in this paragraph why other effects are (at least here) of second order and neglected, e.g., advective transport, rigid body motion, …

We will mention in this paragraph that other processes could play a role in the evolution of SSA. See the response on Specific Remark 1.

Line 38: one key parameter…

We will change **Line 38** accordingly:

*"In this picture, one key parameter driving snow metamorphism is […]"*

Line 61: I think alpha also impacts numerical effort for the simpler models. Thus, suggestion: Since the choice of alpha has..., it is not surprising that…

Line 64: suggestion: at the expense of …

We will change **Line 61** accordingly:

*"Since the choice of α has a significant impact on numerical effort, it is not surprising that the majority of modeling attempts exist for simplified geometries (mostly spheres) (Adams and Brown, 1982; Colbeck, 1983; Albert and McGilvary, 1992; Miller and Adams, 2009), at the expense of microstructural realism."*

Line 65: detailed is somewhat misleading here (in comparison to the mu-CT models with detailed micro-structure). Suggestion to remove "detailed".

We will change **Line 65** accordingly:

*"[…] are those implemented in snow cover models e.g., […]"*

Line 73: it is not entirely clear to me why not both parameters could be measured experimentally (e.g., extracted from 4D mu-CT) or also both be computed by a model. Maybe you could simply say that what you propose is one possibility.

Furthermore, it seems to me that you also compute the interfacial curvature from the model.

What we meant is that when simulating the SSA evolution from a given 3D image, the computation of the curvature and interface velocity fields are fundamentally different. The first one is purely geometric and is readily accessible with the 3D image. The second is not directly given by the 3D image and requires extra knowledge about the physics at play. It is true that time-series of 3D images could be used to experimentally estimate the interface velocity, but our goal in the paper is to be able to compute how the SSA of a given microstructure evolves over time (without the need for information about the microstructure in the future).

We will specify that in the text in **Line 73**:

*"While the interfacial curvature is a purely geometrical quantity that can directly be computed from a µCT image, $v_n$ is a physical quantity that further depends on the involved physical processes."*

Line 80: later, in the discussion, you (I think, correctly) say that your model works on 3D mu-CT images. I would find it useful to be clear/explicit here in the intro: do you actually use a time-series of 3D images and, for each image, calculate SSA-evolution using the model Or - vs. only use the 1st image and compute from there the SSA evolution for the whole future.

For a 3D image at time $t_n$, we compute the growth rate based on its microstructure and then use this growth rate to update our time series of SSA values from $s^n$ to $s^{n+1}$. We will specify it at the start of the Numerical Modeling section **Line 140**, as detailed in the response to the Specific Remark 2.

Line 115: so far, alpha was called vapor attachment coefficient

We will introduce alpha in **Line 38** defining it "condensation coefficient" and mentioning that other names of alpha appear in the literature:

*"In this picture, one key parameter driving snow metamorphism is the condensation coefficient α, also called attachment, kinetic or sticking coefficient (Libbrecht, 2005; Kaempfer & Plapp, 2009;*

*Krol & Loewe, 2016; Demange et al., 2017; K. Fourteau et al., 2021b; L. Bouvet et al., 2022) that controls the kinetics of vapor deposition and sublimation."*

We will further call alpha "condensation coefficient" in abstract (**Line 8**) and throughout the text.

Line 127: hmm... you also use the mean curvature. Maybe say: this information, together with information about surface curvature, is…

We agree and therefore will change **Line 126** accordingly:

*"[…] this information, together with information about surface curvature, is sufficient to […]"*

Line 128: suggestion (see also comment high up, where SSA is defined): define SSA "per unit volume", remove parenthesis here.

Further down, SSA_V is introduced; maybe this could already be done at the definition high up (or there, one could at least say that the normalization can done in several ways).

minor suggestion: the "evolution equation" for SSA…

We will rewrite this subsection by first introducing both specific surface area definitions and then presenting the surface area evolution equation **Line 125**:

*"In this article, we use of two SSA definitions: specific surface area per unit volume s and specific surface area per ice volume $SSA_v$. They are closely related through the ice volume fraction $\varphi_i$:*

*Eq. 8*

*We mainly work with the quantity s for the rest of the article. However, we note that the quantity $SSA_v$ is more commonly used in the snow community (.e.g Matzl and Schneebeli, 2006), since it directly corresponds to the optical diameter.*

*The solution of heat and mass diffusion equations (Eq. (1)-(3)) with boundary conditions (Eq. (4)-(7)) yields […] As a result, for single grains or statistically homogeneous microstructures, the surface area evolution equation can be expressed as follows: […]"*

Line 130: not entirely clear to me: do you average over the same unit volume as for the SSA? Or is it a smaller average?

The surface average is performed on the ice-air interface of a given snow volume. This strictly corresponds to the interface from which the SSA is defined.

Line 134: simplify: "Eq. (9) allows us…"

We will change **Line 134** removing the word "representation":

*"Equation (9) allows us to […]"*

Line 150: shorten: use mu-CT (after first having introduced the abbreviation) throughout.

Line 150: suggestion: taken instead of extracted

We will change **Line 150** accordingly:

*"The µCT image data were taken from [...]"*

We will further change **Line 141**:

*"3.1 µCT time lapse experiments"*

Line 154: shouldn't voxel size be m^3 - or say "in each direction" or "cubic voxel size with side-length"

We will exchange "voxel size" with "voxel side length" in the revised manuscript **Line 154**.

Line 174: I suspect Gamma here to be the "discretized" surface - correct? This is not entirely consistent with the (general) definition of Gamma at the beginning of 2.2. The same comment holds for H. Think about, if it is worth distinguishing.

Indeed, while Gamma in the "Theoretical background" Section corresponds to the "true" ice-air interface in a snow sample, the Gamma of Line 174 corresponds to the triangulated representation. We will change Lines 173-175 accordingly:

*"In addition, we computed the boundary weight on each mesh node k of the triangulated ice-air interface $\Gamma_h$*

$$\omega_k = \int_{\Gamma_h} \psi_k \, d\Gamma_h \quad (13)$$

*where [...]"*

Line 180: suggestion: use consistent terminology (e.g., we solve the diffusion equations (1)-(3) - or alternatively, talk about Laplace eqs higher up).

We will introduce the equations in **Line 94** as (stationary) heat and mass diffusion equations mentioning that they can be called Laplace equations:

*"[...], we employ the common assumption of small particle Péclet number (Libbrecht, 2005) and consider stationary heat and mass diffusion equations (i.e. Laplace equations)."*

Line 181: employing the open-source...

We will change **Line 180** accordingly:

*"On the tetrahedral FE mesh with preserved surface, we solve heat and mass diffusion equations (Eq. (1) - (3)) employing the open-source FE software Elmer (Malinen and Råback, 2013)."*

We will further change to the consistent use throughout the whole text**.**

Line 182: these two steps are not entirely clear to me. I suggest to re-formulate it. My questions are in particular:

- why exactly is one solution of the heat equation not sufficient?

- do you really use temperature gradients as boundary conditions (i.e., flux?) and not Dirichlet temperature b.c.'s?

- is the issue the "vertial" boundary, i.e., the air gap or the boundary condition at the top and bottom?

Due to the presence of artificial air-padding, solving the heat equation once on the given FE mesh (snow image plus air-padding around the image) would result in a smaller temperature gradient

within the snow itself. We aim to obtain a correct temperature gradient within the snow structure (the same as in the experiment). To achieve this, we solve the heat equation on the FE mesh twice: the first one tells us how a gradient over the entire domain translates into the snow itself, and the second uses a corrected gradient to match the experimental gradient in the snow itself. We impose Dirichlet boundary conditions on the top and bottom of the FE mesh to apply the macroscopic temperature gradient. Note that we do not impose a microscopic gradient at the boundaries (which would be a Neumann boundary condition).

Following our response to Specific Remark 2, we will reformulate the manuscript **Line 181** to better explain these two steps.

Line 187: use consistent terminology throughout the paper: heat equation vs. diffusion equation, vs. heat diffusion equation (and similar with vapor)

We will change to the consistent use of heat and mass diffusion equations as stated in response to the comment to Line 180.

Line 207: For my clarification: is Calculate Loads simply providing the flux through an interface? If so, maybe simply say so. And, if you like, add: this can be interpreted as deposition and sublimation fluxes. I am not sure if the simulation does provide directly deposition and sublimation information.

The Elmer function Calculate Loads provides the flux at the interface nodes, expressed in kg/s. It then needs to be converted in surface flux expressed in ks/m$^2$/s. We will change **Line 206** accordingly:

==*"Finally, the required local interface velocity $v_n$ is computed using the vapor flux deduced from the FE simulation. For this, we use the Calculate Loads option of Elmer that provides the vapor flux $f_k$ (expressed in kg s$^{-1}$) at each node $k$ of the ice-air interface. Dividing by the associated boundary weight $\Omega_k$ yields the corresponding deposition/sublimation flux (expressed in kg m$^{-2}$ s$^{-1}$) over the ice-air interface."*==

Line 218: this sentence needs revision (no sense)

We will modify the paragraph for clarity **Line 216**:

==*"For that, we use the VTK package and first cut off the small air padding on the sides using vtkClipDataSet. Then, the triangulated ice-air interface is extracted. The local interface velocity $v_n$ is directly taken from the FE simulation using Eq.(17) . For the local curvature H, we employ the image analysis derived in Krol and Loewe (2018), based on the shape operator, as explained in Section 3.2. Finally, the surface integration for the average in $<v_n H(t_n)>$ takes into account the variable element size of the triangular mesh of the ice-air interface."*==

Line 222: During a 1st read, I had the impression that the error discussion focusses on discretization errors in time.

I would find it useful to clearly state the role of the variance sigma (maybe after having said "of unknown origin") and how it can include dicretization in space or smoothing related errors. Maybe even say already here how one may estimate sigma.

We will specify in the text that these errors could be of whatever origin, for instance due to the variability between the actual ice-air interface, and the reconstructed surface through µCT. **Line 223**:

*"While the combination of the theoretical solution of the diffusion equation and the SSA evolution is, in principle, exact, the 4D image data processing and the derived SSA are subject to experimental and processing errors. These errors could be of various origins, for instance due to uncertainties related to the estimation of the ice-air interface from the µCT scans or to errors related to the numerical FE discretization. When focusing on the temporal evolution of SSA over time, these errors will accumulate and be propagated into the modeled decrease of s(t). To analyze how these errors translates to the overall SSA decrease, and how this depends on the temporal resolution, we resort to a simple stochastic error treatment."*

**Line 223: equations (heat and mass)**

We will change to the consistent use of heat and mass diffusion equations as stated in response to the comment to Line 180.

**Line 225: assess (instead of address)?**

We will change **Line 225** accordingly:

*"[…] error treatment to assess the impact of these errors."*

**Line 227: is there a reason to use t' here and not tau anymore? If not, I suggest to unify the notation**

There is no specific reason for using t'. We will exchange t' with $\tau$ in **Eqs. (19), (20), (22) and Lines 228 and 231**. We will further correct **Eq. (21)** using $\tau$ for all terms in the equation:

*"$r(\tau) = r^{true}(\tau) + \delta r(\tau)$"*

**Line 228: do we really want to introduced "decay rate"? is it not rather "rate of evolution"**

We will change **Lines 72, 130, 134, 216, 228, 231, 363 and 442** determining $v_n H$ as "growth rate" consistently throughout the text.

**Line 233: its rather "computations" than "measurements" that are affected, I think**

Indeed. We will change **Line 233** accordingly:

*"[…], which affects the computations at each time step."*

**Line 254: should this be "and" and not "as" (?)**

We will clarify the sentence **Line 253**:

*"We set up a complex numerical workflow that starts from a voxel image, computes the interface velocity $v_n$ from a FE simulation, and eventually yields the growth rate $<v_n H>$ after surface integration."*

**Line 257: should we say that it is an isothermal situation?**

The vapor problem between the outer and inner shells is actually temperature-independent, rather than isothermal. There is no need for a temperature field in this case. We will revise the paragraph and explain that the problem is temperature-independent **Line 255**:

*"To this end, we employ the classical situation of the Laplace equation in a spherical shell for the vapor concentration $\rho_v(r)$ with radial coordinate r around a spherical particle with radius R with fixed vapor concentration $\rho_\infty$ applied at the outer shell at distance $R_\infty$. A Robin boundary condition (Eq. (15)) is applied at the inner surface of the sphere, under the form $Dv\, n \cdot \nabla\rho_v = \alpha\, v_{kin}[\rho_v - \rho_{v,s}]$, with $\rho_{v,s}$ a constant value smaller than $\rho_\infty$. Note that this problem is temperature-independent and is fully determined by the radius of the shells, and the values $\rho_\infty$ and $\rho_{v,s}$ at the boundaries."*

Figure 1. b) would we not expect v_n to be either positive or negative everywhere?  i.e., why is the scale going from negative to positive values?

From a physical point of view, we expect the vapor field to deposit on the inner sphere and water vapor to be added from the outer sphere to maintain the mass balance. In terms of interface velocity $v_n$, it is positive on the inner sphere (vapor deposition) and negative on the outer sphere (vapor sublimation). We will expand the **caption of Figure 1** specifying that there is sublimation and deposition:

*"Figure 1. [...]  b) Clip of the outer and inner spherical shells with visible elements colored by the interface velocity $v_n$ (sublimation in blue, deposition in red). [...]"*

We will also extent the text **Line 270** to better explain what Fig. 1b illustrates:

"*After solving the vapor equation, with appropriate boundary conditions, we obtain the interface velocity $v_n$, shown in Fig.1b. As expected, we observe a positive velocity on the inner shell, corresponding to vapor deposition, and a negative velocity on the outer shell, corresponding to sublimation.*"

Line 287: please introduce ancronymes when first used.

We will change **Lines 286-287** introducing the acronym:

*"[...] showing the best root mean square error (RMSE) agreement [...]"*

We will further change **caption of Figure 1c** and **Line 303** using only the acronym.

Line 296: does this mean that you use e.g., the 1st image to compute SSA evolution form time 0 to time 48, then the image from time 48 for the next 48h, and so on? See also my remark in the intro-section. I would find it helpful to be more clear / explicit, either above or here in the numerics section.

Yes it is exactly that. We compute the growth rate at a given time $t_n$ using the corresponding microstructure and apply it to the SSA to compute its value at time $t_{n+1}$. We will clarify this in the text, following our response to the Specific Remark 2.

Figure 4: a color scheme that also works for people with color deficiency would be appreciated. Alternatively, use different "dots" or line-styles

We will improve readability of **Figure 4** by changing a color and using dashed lines:

[Figure]

We will change **Line 308** accordingly:

*"[…] for Series 2 is lower despite higher data scattering."*

Indeed, the stochastic model not only incorporates the temporal resolution but also the level of methodological errors. The resulting error on the SSA is the interplay of both. However, the Section mainly focuses on how varying the temporal resolution (while assuming a constant level of methodological errors) impact the modeled SSA decrease. Therefore, we would prefer to keep the title of the Section as such.

It is true that once the stochastic model is adjusted to the simulated data, we obtain a variance that *in principle* characterizes the level of stochastic methodological errors. However, before further interpreting this adjusted variance value, it would likely be beneficial to quantify the robustness of the method and to ensure that this adjusted value is indeed indicative of the overall methodological error in the simulation workflow. This is an interesting prospect for future work.

We will change **Line 310** accordingly:

*"[…] we performed simulations with a time step refined down to the time interval between […]."*

We will change **Line 321** accordingly:

*"On the one hand, this […]"*

Line 367: suggestion: "nevertheless" instead of "as a result of the numerical effort"

We will change **Line 367** accordingly:

*"Nevertheless, we were able to [...]"*

Line 408: Indeed, I think it is a remarkable result that a somehow volumetrically averaged kinetic coefficient seems to be sufficient to explain SSA evolution. The message is here, but not very "pointy".

We will draw attention to this result by adding the following sentence to **Line 405**:

*"It is quite remarkable that despite variations of the condensation coefficient at the micro-scale, their collective behavior can be appropriately described through the use of a single α value. Indeed, in principle, the assumption of a constant α [...]"*

Line 420: how come the volume for the simulation impacts the experimental parameters? Can this be reformulated?

If the volume are quite close or smaller than the REV size, one could observe fluctuations in the SSA values that are due to changes in the observed Region Of Interest rather than variations due to actual evolution of the macroscopic SSA. This will be reformulated in the text **Line 418**:

*"Second, the volume of interest considered here for the simulations is relatively small, in particular for Series 2. This might lead to some non-representativeness issues and fluctuations in the measured SSA. This could explain the noisy nature of the experimental parameter curves."*

Line 425: yes! I think, you should not entitle section 3 as "temporal" only and have the discussion along the line of this nice summary.

Following the Specific Remark 3, we will mention earlier in the manuscript **Line 223** that the stochastic model incorporates the interplay between temporal resolution and methodological uncertainties.

Line 434: I am not sure that non-convergence of the solver is linked to this simplification. I would keep the two discussions separate. You could e.g., justify (higher up, where it is first said) to neglect latend heat for simplicity with the argument to increase numerical stability. Then, in a separate paragraph under 5.4. say that numerical improvements might help to increase numerical convergence.

This simplification is a trade-off between numerical simplicity and physical realism. It helps the convergence of the solver, as the numerical problems to be solved are smaller and less complex, but remove a potentially important physical process.

We will explain early in the revised manuscript why latent heat is neglected **Line 108**:

*"As by Krol and Löwe (2016), the latent heat during the sublimation and deposition is neglected for reduced model complexity."*

We will clarify why this simplification helps convergence, but that despite it there are still converge issues, and propose so remedy for it **Line 432**:

*"This leads to a slightly simpler numerical situation where heat and vapor are coupled only one way, and the heat diffusion equation can be solved in advance. This strategy reduces the numerical cost of the method and facilitate the convergence of iterative solver used in the FE software. Despite this simplification, we still observe that the vapor solver had issues to converge for a few microstructures, which explains a few missing points in the modeled time series (e.g., Fig. 4. The convergence of the FE simulations depends on the employed mesh and on the value of α. It could be facilitated by improving the mesh quality or increasing the maximum number of iterations."*

Line 436: It seems not correct that latent heat can contribute to mass fluxes. It might possibly impact them. Please reformulate.

We will rephrase the sentence in **Line 435**:

*"While this one-way coupling assumption eases the numerics, it was previously shown Fourteau et al. (2021) that for low density or fast kinetics, latent heat significantly contributes to the heat fluxes in snow and may thus likewise impact the volume averaged rate term $v_n H$."*

Line 446: I think this such a central result / conclusion that it is worth cross-checking consistency with terminology and already higher up in the document introduce the "effective kinetic coefficient", I think it is to be understood as a volumetric average.

We will introduce the notion of an "effective condensation coefficient" early in the revised manuscript in Sect. 2, **Line 123**:

*"Although α is known to depend on temperature, supersaturation, crystallographic orientation and to vary on different parts of the ice-air interface (Libbrecht ,2005), we rely on the simplifying assumption of a single and constant \alpha value. It should thus rather be understood as an effective condensation coefficient."*

**ADDITIONAL REFERENCES:**

Demange, G., Zapolsky, H., Patte, R., and Brunel, M.: A phase field model for snow crystal growth in three dimensions, npj Computational Materials, 3, 1–7, 2017.

Ebner, P. P., Schneebeli, M., and Steinfeld, A.: Metamorphism during temperature gradient with undersaturated advective airflow in a snow sample, The Cryosphere, 10, 791–797, https://doi.org/10.5194/tc-10-791-2016, 2016.

Schleef, S., Löwe, H., and Schneebeli, M.: Influence of stress, temperature and crystal morphology on isothermal densification and specific surface area decrease of new snow, The Cryosphere, 8, 1825–1838, https://doi.org/10.5194/tc-8-1825-2014, 2014.

---

## Author Comment (AC2)

**Response to RC1 from Z.R. Courville on egusphere-2023-1947**

We are thankful to Z.R. Courville for thorough and constructive review of our manuscript.

We have copied the comments of Z.R. Courville in blue. Our corresponding responses are available in black below each comment and proposed modifications to the manuscripts are written in yellow highlighted italics. The bold text line numbers correspond to the original manuscript.

Best regards

Anna Braun on behalf of all co-authors

The manuscript presents a very interesting physically-based modeling approach to the evolution of snow specific area during temperature gradient metamorphism, notably presenting results constraining the kinetic attachment coefficient, which is difficult to measure and has proved an elusive parameter in overall efforts at understanding temperature gradient metamorphism. I found the result that the kinetic coefficient varied between the two samples, and then over the course of one of the experiments particularly interesting, but intuitively makes sense in terms of the dependence of the kinetic coefficient on the morphology of snow grains. I also find the SSA modeling results presented in Fig 6 compelling with respect to the microCT data, with the match between model and experimental results remarkable. The manuscript is very well written. Below, I offer a few minor suggestions for consideration to improve clarity for a reader. The main suggestion I have is to use a consistent definition of alpha throughout the text. I also had a few questions about the specifics of the model mentioned that I think might warrant clarification.

Thanks a lot for the positive feedback. We are encouraged to confirm the intuitive understanding of temperature gradient metamorphism by a rigorous numerical approach. As detailed below, we will improve the naming of alpha, consistently calling it "condensation coefficient".

Line 64: "at the downside" is not quite the right phrase, I would suggest "at the expense" instead

We will change **Line 64** accordingly:

*"[…], at the expense of microstructural realism."*

Line 73: I would suggest writing: "While the interfacial curvature is a geometrical quantity, the interface growth velocity must be computed from a physical model." That is only a suggestion to make that sentence clearer vs. "first" and "second" term since that sentence has a lot of terms in it, and that's if I've interpreted the sentence correctly.

The sentence will be rewritten in the revised manuscript, removing the words "first" and "second". We also modified the sentence following a comment from Thomas Kaempfer. We will rephrase the text **Line 73** to:

*"While the interfacial curvature is a purely geometrical quantity that can directly be computed from a µCT image, $v_n$ is a physical quantity that further depends on the involved physical processes."*

Line 88: I'm not sure "motion" is the best term for the interface, and would suggest "evolution" or maybe "migration" instead.

We propose to change **Lines 87-88** using the term "evolution":

*"For an arbitrary snow structure, morphological changes during metamorphism are predominantly driven by the coupled diffusion of heat and mass together with ice-air interface evolution due to deposition and sublimation of vapor."*

We will further rephrase "motion of the interface" in **Lines 92-93**:

*"Due to the separation of time scales between heat and mass diffusion in the pores and the evolution of the interface due to crystal growth, [...]"*

Line 89: How was the size of the representative snow volume determined? (or is that in the Pinzer article? If they do discuss how the representative volume was determined, I would mention that briefly.)

Following the reviews, we have performed additional simulations determining a representative elementary volume with respect to the growth rate.

In the figure below, the sample sizes used in the article correspond to the largest volumes for each sample. This shows that the estimated growth rates using these sample sizes are representative.

[Figure]

We will discuss this result, starting **Line 179**:

*"The FE meshes of this article are based on the whole available µCT images. We verified that these selected volumes were large enough to yield representative results.. By varying sub-volumes extracted from the center of µCT images at the start and the end of both series ($I(t_1)$, $I(t_{49})$, $\hat{I}(t_1)$ and $\hat{I}(t_{83})$), we found that the simulated growth rate corresponds to a representative value for the sample sizes used in this study. This is consistent with the results of Calonne et al. (2011) for thermal conductivity, that report representativeness for sample side-lengths between 2.5 and 5mm."*

Line 115: Throughout the text, there are several definitions/names of the parameter alpha (or at least I think they are all referring to alpha). As a suggestion, I recommend either being more consistent, or explaining at the first instance that alpha has been called different things. The first time it happened, I was wondering why the change from "vapor attachment coefficient" to

"condensation coefficient", and recommended defining alpha as "the vapor attachment coefficient, or the condensation coefficient" at the first definition of alpha, but then I noted that there are several different forms of the definition used throughout the manuscript, including "kinetic coefficient" (line 293) and "attachment kinetics coefficient" (line 297). Again, I **think** these are all referring to alpha, but I am not sure.

We will introduce alpha in **Lines 39** and refer to it as the "condensation coefficient". We will also mention that other names of alpha appear in the literature:

*"In this picture, one key parameter driving snow metamorphism is the condensation coefficient α, also called attachment, kinetic or sticking coefficient (Libbrecht, 2005; Kaempfer & Plapp, 2009; Krol & Loewe, 2016; Demange et al., 2017b; K. Fourteau et al., 2021a; L. Bouvet et al., 2022) that controls the kinetics of vapor deposition and sublimation."*

We will further call alpha "condensation coefficient" in abstract (**Line 8**) and throughout the text.

Line 119: Ditto that last comment for the definition of alpha in this instance (I stopped noting all the different terms used for alpha as I went on in my review, see the above comment. I think either calling it the same thing or discussing all the different variations is warranted to alleviate confusion.)

We will change it according to the response to the previous comment.

Line 119: Suggest rewriting as "the kinetic coefficient α is defined as the probability of a water molecule sticking to an impinging surface." (this is only a minor grammar/usage suggestion)

From what we understand, the word impinging applies to the molecule rather than to the surface. We propose to rephrase **Line 119** as:

*"In the Hertz-Knudsen equation, the condensation coefficient α is defined as the probability of a water molecule sticking to a surface after impinging on it."*

Line 123: Is (7) referring to a reference in bibtex or some other citation managing software? Or is it referring to equation 7? Might be clearer if it said "eq. 7"

This should be a reference to Eq. 7. We will change **Line 123** accordingly:

*"[…] as deviations from the local constitutive behavior (Eq. (7)) due to non-local surface processes (Libbrecht, 2005)."*

Line 145: By "shorter" does that mean the sample is physically smaller, or that the time was shorter (I mean, I think I know the answer since the hours are greater for Series 2)? Suggest rewriting to clarify, maybe "Series 1 lasted 384 h and had a shorter sample height…" if that is what is meant. Also seems like the sample thicknesses/heights should be included as a well as the temperature gradients, even if the details are in the Pinzer paper.

No, just the duration is shorter. We will change **Line 145** to avoid confusion:

*"Series 1 lasted 384 h, while Series 2 lasted 665 h."*

We will further correct and rephrase **Line 155**, including the size of the samples:

*"This corresponds to samples of 7.5x7.5x4.9mm$^3$ for series 1 and  5.4.x5.4x3.5mm$^3$ for series 2."*

Line 146: Does mean T refer to the average air/ambient temperature for the experiment or the average temp throughout the sample?

Here, we mean the mean temperature throughout the sample. We will change **Line 146** accordingly:

*"The mean temperature T of the sample [...]"*

Line 159: How was "a reasonable volumetric division" determined or quantified? Specify the requirements.

The main requirement of the mesh beside preserving the surfaces is the achievement of the accurate discretized numerical solution. There is no clear cut-off value here, apart from the general idea that the elements need to be small compared to the length-scale of the physical problem to be solved and that smaller elements *usually* yields less errors. A further constraint of the element size is the available computational power, as smaller elements means more elements.

We will rephrase **Lines 159-160** to be more specific:

*"The production of an appropriate mesh that discretizes the air and ice domains, preserves the ice-air interface, and is fine enough to get accurate numerical solution (without overloading computational resources) is a key requirement for our problem."*

To ensure that we used a sufficient degree of refinement, we have performed additional simulations with different degrees of refinement. This provides information about the sensitivity of our results to the FE mesh. As can be seen from the graph below (stars represent the number of elements that were used in the manuscript), the growth rate $v_nH$ is only reasonably impacted when increasing the number of elements. Moreover, as discussed later, the very good agreement between a FE simulation and the analytical solution for a spherical problem suggests that our meshing criteria yields an appropriate mesh. This will be stated **Line 170**:

*"We have estimated the sensitivity of our results to the FE mesh. We found that doubling the number of elements in the mesh impacted the growth rate by about 10%. This is small in light of the dependence of the SSA values on the condensation coefficient α investigated in this study. Moreover, the very good agreement between a FE simulation and the analytical solution for a spherical problem (see Sect. 3.6) suggests that our meshing criteria yield an appropriate mesh."*

[Figure]

Line 165: Likewise, define "small air padding" quantitatively, or if dependent on the size of the volume of interest/SSA or sample grain size, describe how that was determined.

The thickness of the air-padding layer is 3 voxels around the 300x300x196 snow image. This will be specified in the text **Line 160**:

*"To this end, we employ the open-source Computational Geometry Algorithms Library (CGAL) (The CGAL Project, 2022). Specifically, we use the class Polyhedral_mesh_domain_with_features_3 that implements a tetrahedral mesh of a domain bounded by polyhedral surfaces that are preserved. The provided surfaces need to be closed and free of self-intersections. To obtain such surfaces, we extract the ice-air interface from the binary µCT data (Eq. (11) and (12)) following the procedure from (Krol and Löwe, 2018), namely by applying a Gaussian smoothing and the contour filter from the Visualization Toolkit (VTK) (Schroeder et al., 2006). However, by default this procedure applied to µCT images yields a surface that is open at the boundaries of the domain. In order to obtain closed surfaces, we added a small air-padding (three voxel-thick) around the image. This allowed us to properly define a closed outer boundary suitable for meshing. As detailed below, we provided special care to ensure that the introduction of this artificial air-padding does not perturb the simulation within the snow microstructure itself."*

Line 189: For readers not familiar with Elmer, it would be good to add a brief description of what an ILU preconditioner is or does. I will note, though, that in general the authors have done a very good job of describing what the different functions in Elmer are for a non-Elmer user.

We will add a reference for the ILU preconditioner in **Line 188**:

*"The equations are solved with the iterative biconjugate gradient stabilized method (BiCGSTAB; Van der Vorst, 1992) together with an ILU preconditioner, meant to facilitate the numerical solving by performing an incomplete LU factorization (Saad, 1996)."*

Line 263: I would put in the length scale of the test case (0.9 mm) so the reader doesn't have to do the math, i.e., "In this way, the length scales of the test case (0.9 mm for the outer radius) are a similar order of magnitude…"

We add the lengths of the inner and outer radii **Line 264**:

"where the inner radius is set to R=21 voxel and the outer radius set to R_{\infty}=51 voxel with a voxel size of 18µm, corresponding to inner and outer radii of 0.38 and 0.92mm, respectively."

Figure 1. Suggest putting a scale bar in for the sphere (in mm) if it doesn't clutter the figure too much since that will help a reader compare to typical snow grain sizes, or adding the outer sphere dimension to the caption.

We will add a scale bar to the Figure 1b. The revised Figure is displayed below.

Figure 1. For b) is the blue the "air padding" similar to what was added to the microCT volume?

The blue color in the Figure 1b does not correspond to an air-padding layer but to the outer sphere. It is blue as vapor sublimates from the outer sphere. We will state that more clearly in the caption of the Figure 1:

*"Figure 1. [...] b) Clip of the outer and inner spherical shells with visible elements colored by the interface velocity $v_n$ (sublimation in blue, deposition in red)."*

Moreover, we will extent the text in **Lines 270** to discuss where sublimation and deposition occur:

*"After solving the vapor equation, with appropriate boundary conditions, we obtain the interface velocity $v_n$, shown in Fig.1b. As expected, we observe a positive velocity on the inner shell, corresponding to vapor deposition, and a negative velocity on the outer shell, corresponding to sublimation."*

Figure 1. For c) what are the red and blue dashed lines showing? I'm guessing that is the (sim-theo)/theo for values of alpha, but that should be called out in the legend, and which axis those values are plotted on should be indicated for easy of reading.

Indeed, the red and blue dashed lines corresponds to the relative errors.

This will be mentioned in the caption of **Fig. 1**:

*"Figure 1. [...] c) Comparison of the growth rate $v_n H$ on the inner radius R of theoretical (theo) and simulated (sim) solution of the spherical shell test case for different values of the condensation coefficient α. Two different surface mesh qualities with (smooth) and without (non-smooth) smoothing are employed. The red dots, blue squares and black solid line correspond to $v_n H$ on the left y-axis while the dashed red and blue lines correspond to simulation error on the right y-axis."*

[Figure]

Line 308: what does the RMSE minimum "is deeper" mean? That the RMSE minimum is lower?

We will replace "deeper" with "lower" **Line 308** to:

*"[…] the RMSE minimum for Series 2 is lower despite higher data scattering."*

Line 311: Should be "a time step refined down to the time interval between two microCT images…" or something (seems like there is a missing preposition after "down").

We will change **Line 311** accordingly:

*"[…] we performed simulations with a time step refined down to the time interval between […]."*

**ADDITIONAL REFERENCES:**

Demange, G., Zapolsky, H., Patte, R., & Brunel, M.: A phase field model for snow crystal growth in three dimensions, Npj Computational Materials, 3(1), 1–7, https://doi.org/10.1038/s41524-017-0015-1, 2017b.

Saad, Y.: Iterative Methods for Sparse Linear Systems, Society for Industrial and Applied Mathematics, 2003.

---

## Referee Report (RR1)

Dear Editor,

Please find the revised manuscript of the research article egusphere-2023-1947.

In this document, we have attached our responses to the two reviews. The track-changes version of the manuscript is available at the end. Added text is shown in blue underlined, while removed text in shown in red strikeout.

Please also note, that we have corrected the values of inner and outer radii used in Section 3.5. Indeed, we only realized after submitting the response to the referees that the numbers initially given in the manuscript did not exactly correspond to those used in the simulations and in Figure 1. This modification does not change the results or conclusions of Section 3.5 on the validation of the workflow.

Kind regards
Anna Braun on behalf of all co-authors

**Response to RC1 from Z.R. Courville on egusphere-2023-1947**

We are thankful to Z.R. Courville for thorough and constructive review of our manuscript.

We have copied the comments of Z.R. Courville in blue. Our corresponding responses are available in black below each comment and proposed modifications to the manuscripts are written in yellow highlighted italics. The bold text line numbers correspond to the original manuscript.

Best regards

Anna Braun on behalf of all co-authors

The manuscript presents a very interesting physically-based modeling approach to the evolution of snow specific area during temperature gradient metamorphism, notably presenting results constraining the kinetic attachment coefficient, which is difficult to measure and has proved an elusive parameter in overall efforts at understanding temperature gradient metamorphism. I found the result that the kinetic coefficient varied between the two samples, and then over the course of one of the experiments particularly interesting, but intuitively makes sense in terms of the dependence of the kinetic coefficient on the morphology of snow grains. I also find the SSA modeling results presented in Fig 6 compelling with respect to the microCT data, with the match between model and experimental results remarkable. The manuscript is very well written. Below, I offer a few minor suggestions for consideration to improve clarity for a reader. The main suggestion I have is to use a consistent definition of alpha throughout the text. I also had a few questions about the specifics of the model mentioned that I think might warrant clarification.

Thanks a lot for the positive feedback. We are encouraged to confirm the intuitive understanding of temperature gradient metamorphism by a rigorous numerical approach. As detailed below, we will improve the naming of alpha, consistently calling it "condensation coefficient".

Line 64: "at the downside" is not quite the right phrase, I would suggest "at the expense" instead

We will change **Line 64** accordingly:

*"[…], at the expense of microstructural realism."*

Line 73: I would suggest writing: "While the interfacial curvature is a geometrical quantity, the interface growth velocity must be computed from a physical model." That is only a suggestion to make that sentence clearer vs. "first" and "second" term since that sentence has a lot of terms in it, and that's if I've interpreted the sentence correctly.

The sentence will be rewritten in the revised manuscript, removing the words "first" and "second". We also modified the sentence following a comment from Thomas Kaempfer. We will rephrase the text **Line 73** to:

*"While the interfacial curvature is a purely geometrical quantity that can directly be computed from a μCT image, $v_n$ is a physical quantity that further depends on the involved physical processes."*

Line 88: I'm not sure "motion" is the best term for the interface, and would suggest "evolution" or maybe "migration" instead.

We propose to change **Lines 87-88** using the term "evolution":

*"For an arbitrary snow structure, morphological changes during metamorphism are predominantly driven by the coupled diffusion of heat and mass together with ice-air interface evolution due to deposition and sublimation of vapor."*

We will further rephrase "motion of the interface" in **Lines 92-93**:

*"Due to the separation of time scales between heat and mass diffusion in the pores and the evolution of the interface due to crystal growth, [...]"*

Line 89: How was the size of the representative snow volume determined? (or is that in the Pinzer article? If they do discuss how the representative volume was determined, I would mention that briefly.)

Following the reviews, we have performed additional simulations determining a representative elementary volume with respect to the growth rate.

In the figure below, the sample sizes used in the article correspond to the largest volumes for each sample. This shows that the estimated growth rates using these sample sizes are representative.

[Figure]

We will discuss this result, starting **Line 179**:

*"The FE meshes of this article are based on the whole available µCT images. We verified that these selected volumes were large enough to yield representative results.. By varying sub-volumes extracted from the center of µCT images at the start and the end of both series (I(t₁), I(t₄₉), Î(t₁) and Î(t₈₃)), we found that the simulated growth rate corresponds to a representative value for the sample sizes used in this study. This is consistent with the results of Calonne et al. (2011) for thermal conductivity, that report representativeness for sample side-lengths between 2.5 and 5mm."*

Line 115: Throughout the text, there are several definitions/names of the parameter alpha (or at least I think they are all referring to alpha). As a suggestion, I recommend either being more consistent, or explaining at the first instance that alpha has been called different things. The first time it happened, I was wondering why the change from "vapor attachment coefficient" to

"condensation coefficient", and recommended defining alpha as "the vapor attachment coefficient, or the condensation coefficient" at the first definition of alpha, but then I noted that there are several different forms of the definition used throughout the manuscript, including "kinetic coefficient" (line 293) and "attachment kinetics coefficient" (line 297). Again, I **think** these are all referring to alpha, but I am not sure.

We will introduce alpha in **Lines 39** and refer to it as the "condensation coefficient". We will also mention that other names of alpha appear in the literature:

*"In this picture, one key parameter driving snow metamorphism is the condensation coefficient α, also called attachment, kinetic or sticking coefficient (Libbrecht, 2005; Kaempfer & Plapp, 2009; Krol & Loewe, 2016; Demange et al., 2017b; K. Fourteau et al., 2021a; L. Bouvet et al., 2022) that controls the kinetics of vapor deposition and sublimation."*

We will further call alpha "condensation coefficient" in abstract (**Line 8**) and throughout the text.

Line 119: Ditto that last comment for the definition of alpha in this instance (I stopped noting all the different terms used for alpha as I went on in my review, see the above comment. I think either calling it the same thing or discussing all the different variations is warranted to alleviate confusion.)

We will change it according to the response to the previous comment.

Line 119: Suggest rewriting as "the kinetic coefficient α is defined as the probability of a water molecule sticking to an impinging surface." (this is only a minor grammar/usage suggestion)

From what we understand, the word impinging applies to the molecule rather than to the surface. We propose to rephrase **Line 119** as:

*"In the Hertz-Knudsen equation, the condensation coefficient α is defined as the probability of a water molecule sticking to a surface after impinging on it."*

Line 123: Is (7) referring to a reference in bibtex or some other citation managing software? Or is it referring to equation 7? Might be clearer if it said "eq. 7"

This should be a reference to Eq. 7. We will change **Line 123** accordingly:

*"[…] as deviations from the local constitutive behavior (Eq. (7)) due to non-local surface processes (Libbrecht, 2005)."*

Line 145: By "shorter" does that mean the sample is physically smaller, or that the time was shorter (I mean, I think I know the answer since the hours are greater for Series 2)? Suggest rewriting to clarify, maybe "Series 1 lasted 384 h and had a shorter sample height…" if that is what is meant. Also seems like the sample thicknesses/heights should be included as a well as the temperature gradients, even if the details are in the Pinzer paper.

No, just the duration is shorter. We will change **Line 145** to avoid confusion:

*"Series 1 lasted 384 h, while Series 2 lasted 665 h."*

We will further correct and rephrase **Line 155**, including the size of the samples:

*"This corresponds to samples of 7.5x7.5x4.9mm$^3$ for series 1 and  5.4.x5.4x3.5mm$^3$ for series 2."*

Line 146: Does mean T refer to the average air/ambient temperature for the experiment or the average temp throughout the sample?

Here, we mean the mean temperature throughout the sample. We will change **Line 146** accordingly:

*"The mean temperature T of the sample [...]"*

Line 159: How was "a reasonable volumetric division" determined or quantified? Specify the requirements.

The main requirement of the mesh beside preserving the surfaces is the achievement of the accurate discretized numerical solution. There is no clear cut-off value here, apart from the general idea that the elements need to be small compared to the length-scale of the physical problem to be solved and that smaller elements *usually* yields less errors. A further constraint of the element size is the available computational power, as smaller elements means more elements.

We will rephrase **Lines 159-160** to be more specific:

*"The production of an appropriate mesh that discretizes the air and ice domains, preserves the ice-air interface, and is fine enough to get accurate numerical solution (without overloading computational resources) is a key requirement for our problem."*

To ensure that we used a sufficient degree of refinement, we have performed additional simulations with different degrees of refinement. This provides information about the sensitivity of our results to the FE mesh. As can be seen from the graph below (stars represent the number of elements that were used in the manuscript), the growth rate $v_nH$ is only reasonably impacted when increasing the number of elements. Moreover, as discussed later, the very good agreement between a FE simulation and the analytical solution for a spherical problem suggests that our meshing criteria yields an appropriate mesh. This will be stated **Line 170**:

*"We have estimated the sensitivity of our results to the FE mesh. We found that doubling the number of elements in the mesh impacted the growth rate by about 10%. This is small in light of the dependence of the SSA values on the condensation coefficient α investigated in this study. Moreover, the very good agreement between a FE simulation and the analytical solution for a spherical problem (see Sect. 3.6) suggests that our meshing criteria yield an appropriate mesh."*

[Figure]

 Likewise, define "small air padding" quantitatively, or if dependent on the size of the volume of interest/SSA or sample grain size, describe how that was determined.

The thickness of the air-padding layer is 3 voxels around the 300x300x196 snow image. This will be specified in the text **Line 160**:

*"To this end, we employ the open-source Computational Geometry Algorithms Library (CGAL) (The CGAL Project, 2022). Specifically, we use the class Polyhedral_mesh_domain_with_features_3 that implements a tetrahedral mesh of a domain bounded by polyhedral surfaces that are preserved. The provided surfaces need to be closed and free of self-intersections. To obtain such surfaces, we extract the ice-air interface from the binary µCT data (Eq. (11) and (12)) following the procedure from (Krol and Löwe, 2018), namely by applying a Gaussian smoothing and the contour filter from the Visualization Toolkit (VTK) (Schroeder et al., 2006). However, by default this procedure applied to µCT images yields a surface that is open at the boundaries of the domain. In order to obtain closed surfaces, we added a small air-padding (three voxel-thick) around the image. This allowed us to properly define a closed outer boundary suitable for meshing. As detailed below, we provided special care to ensure that the introduction of this artificial air-padding does not perturb the simulation within the snow microstructure itself."*

 For readers not familiar with Elmer, it would be good to add a brief description of what an ILU preconditioner is or does. I will note, though, that in general the authors have done a very good job of describing what the different functions in Elmer are for a non-Elmer user.

We will add a reference for the ILU preconditioner in **Line 188**:

*"The equations are solved with the iterative biconjugate gradient stabilized method (BiCGSTAB; Van der Vorst, 1992) together with an ILU preconditioner, meant to facilitate the numerical solving by performing an incomplete LU factorization (Saad, 1996)."*

 I would put in the length scale of the test case (0.9 mm) so the reader doesn't have to do the math, i.e., "In this way, the length scales of the test case (0.9 mm for the outer radius) are a similar order of magnitude…"

We add the lengths of the inner and outer radii **Line 264**:

"where the inner radius is set to R=21 voxel and the outer radius set to R_{\infty}=51 voxel with a voxel size of 18µm, corresponding to inner and outer radii of 0.38 and 0.92mm, respectively."

Figure 1. Suggest putting a scale bar in for the sphere (in mm) if it doesn't clutter the figure too much since that will help a reader compare to typical snow grain sizes, or adding the outer sphere dimension to the caption.

We will add a scale bar to the Figure 1b. The revised Figure is displayed below.

Figure 1. For b) is the blue the "air padding" similar to what was added to the microCT volume?

The blue color in the Figure 1b does not correspond to an air-padding layer but to the outer sphere. It is blue as vapor sublimates from the outer sphere. We will state that more clearly in the caption of the Figure 1:

*"Figure 1. [...] b) Clip of the outer and inner spherical shells with visible elements colored by the interface velocity $v_n$ (sublimation in blue, deposition in red)."*

Moreover, we will extent the text in **Lines 270** to discuss where sublimation and deposition occur:

*"After solving the vapor equation, with appropriate boundary conditions, we obtain the interface velocity $v_n$, shown in Fig.1b. As expected, we observe a positive velocity on the inner shell, corresponding to vapor deposition, and a negative velocity on the outer shell, corresponding to sublimation."*

Figure 1. For c) what are the red and blue dashed lines showing? I'm guessing that is the (sim-theo)/theo for values of alpha, but that should be called out in the legend, and which axis those values are plotted on should be indicated for easy of reading.

Indeed, the red and blue dashed lines corresponds to the relative errors.

This will be mentioned in the caption of **Fig. 1**:

*"Figure 1. [...] c) Comparison of the growth rate $v_nH$ on the inner radius R of theoretical (theo) and simulated (sim) solution of the spherical shell test case for different values of the condensation coefficient α. Two different surface mesh qualities with (smooth) and without (non-smooth) smoothing are employed. The red dots, blue squares and black solid line correspond to $v_nH$ on the left y-axis while the dashed red and blue lines correspond to simulation error on the right y-axis."*

[Figure]

Line 308: what does the RMSE minimum "is deeper" mean? That the RMSE minimum is lower?

We will replace "deeper" with "lower" **Line 308** to:

*"[...] the RMSE minimum for Series 2 is lower despite higher data scattering."*

Line 311: Should be "a time step refined down to the time interval between two microCT images…" or something (seems like there is a missing preposition after "down").

We will change **Line 311** accordingly:

*"[...] we performed simulations with a time step refined down to the time interval between [...]."*

**ADDITIONAL REFERENCES:**

Demange, G., Zapolsky, H., Patte, R., & Brunel, M.: A phase field model for snow crystal growth in three dimensions, Npj Computational Materials, 3(1), 1–7, https://doi.org/10.1038/s41524-017-0015-1, 2017b.

Saad, Y.: Iterative Methods for Sparse Linear Systems, Society for Industrial and Applied Mathematics, 2003.

**Response to RC2 from Thomas Kaempfer on egusphere-2023-1947**

We are thankful to Thomas Kaempfer for thorough and constructive review of our manuscript.

We have copied the comments from annotated PDF of the review, with the corresponding text line numbers, below in blue. Our responses are available in black below the comments. Proposed modifications to the manuscript are given in yellow highlighted italics. The bold text line numbers correspond to the original manuscript.

Best regards

Anna Braun on behalf of all co-authors

The paper presents a novel approach to model the evolution of the specific surface area (SSA) during temperature gradient metamorphism (TGM). It uses X-ray micro-computed tomography (mu-CT) images of snow in combination with a numerical solution of steady-state energy and mass conservation equations at the micro-structural scale and a surface area equation based on physics first principles. The only "free" parameter in the model is the vapor attachment coefficient alpha and it is proposed that SSA evolution can be predicted using an adequately chosen "effective" alpha.

The paper is generally well written with a strong emphasis on the numerical solution and analysis of error propagation. The strength and limitations of the approach are clearly presented.

Clarity could be improved by using more concise language and unified terminology (see minor comments in annotated PDF-file). The paper could further benefit by considering the following specific remarks. I suggest the paper to be revised accordingly before accepting it for publication.

**Specific remarks**

1. Introduction, line 36ff: While it is OK to quickly concentrate on the relevant mechanisms for the TGM situation studied in this paper (energy and mass conservation, attachment kinetics), I suggest justifying why other processes are of second order instead of "boldly" postulating that it is all about alpha.

It is indeed true that while heat and vapor diffusion together with vapor attachment kinetics are usually the processes used to study snow metamorphism from the pore scale, other processes (such a mechanical deformation or air advection) might interact metamorphism and the temporal evolution of the SSA. This will be clarified in the introduction by adding text **Line 36:**:

*"Physical models of snow metamorphism must comply with the ice crystal growth dynamics at the pore scale (Krol and Löwe, 2016), which includes heat and vapor diffusion, accommodated by attachment kinetics controlling the deposition and sublimation of water molecules onto the ice lattice (Colbeck, 1983; Libbrecht, 2005). Secondary effects on the temporal SSA evolution might be expected from other processes like mechanical deformation (Wang and Backer, 2013; Schleef et al., 2014), advection of air in the porosity (Ebner et al., 2016, Jafari et al., 2022). In this picture, one key parameter driving snow metamorphism is [...]".*

We also propose to change the first paragraph of Sect. 2.1, **Lines 87-94** and explain that our motivation to neglect the potential influence of mechanics and air movement is twofold: (i) it corresponds to the basic mechanisms used to explain metamorphism in the most of the literature and is thus a good candidate in terms of minimum-required complexity, and (ii) it is consistent with the set-up under which the experimental data were acquired:

*"For an arbitrary snow structure, morphological changes during metamorphism are predominantly driven by the coupled diffusion of heat and mass together with ice-air interface evolution due to deposition and sublimation of vapor. In the following, we closely follow the descriptions by Kaempfer et al., (2009), Calonne et al., (2014), Krol and Loewe (2016), and Fourteau et al. (2020). We consider a representative snow volume at the micro-scale consisting of ice and air and denote the sub-domains occupied by the ice and air phase by $\Omega_i$ and $\Omega_a$, respectively. In the following, subscripts i and a denote quantities which are defined in the respective domains $\Omega_i$ and $\Omega_a$. Due to the separation of time scales between heat and mass diffusion in the pores and the evolution of the interface due to crystal growth, we employ the common assumption of small particle Péclet number (Libbrecht, 2005) and consider stationary heat and mass diffusion equations. Furthermore, we neglect the influence of mechanical deformation, as usually done in pore-scale metamorphism models (e.g., (Calonne et al., 2014b; Krol and Löwe, 2016)). We also neglect the potential presence of convection and air advection in the pore space. These assumptions are consistent with the experimental data used here, obtained under controlled laboratory conditions (Pinzer et al., 2012). They are also good candidates in terms of minimum-required complexity to model SSA evolution from pore-scale physics. The partial density of water vapor [...]".*

2.Presentation of the approach (e.g., Introduction, lines 70ff; or beginning of section 3, lines 159ff – possibly add a new sub-subsection at the beginning of sub-section 3.2): The coupling of the mu-CT images and the numerical models (Finite Elements and surface equation) should somewhere carefully be explained. If I understood it correctly, you have resp. do:

- time-laps mu-CT images of snow (4D)
- use a (sub-)set of 3D images as input to your numerical model
- for each chosen 3D image, pre-process and discretize (e.g., image processing, triangulation)
- determine some parameters from the geometry directly (e.g., surface curvature)
- determine other parameters from numerical modeling (e.g., growth velocity)
- compute SSA evolution using above; fit alpha to the experimental results

Yes, your description of the workflow is essentially correct. Specifically we:

1- Select a µCT time series with a given time resolution

2- Initialize the first term $s^1$ of the simulated SSA time series with the value deduced from the first µCT scan.

3- For each timestep $t_n$ of the time series, we compute the SSA growth rate associated with the corresponding microstructure (using a FE simulations).

4- Use the growth rate to prolong the SSA time series from $s^n$ to $s^{n+1}$.

A schematic summarizing this workflow is shown below.

[Figure]

- Unclear to me are in particular:

- how many 3D images do you use for the modeling? When and how do you decide to "update" the micro-structure in your models (e.g., using a new image from the 4D series)? Or do you never do this (and always use the 1$^{st}$ image only)? In general, the discretization in time of the numerical model is unclear to me.

In the modeling, we use one 3D image for each timestep of a time series. As detailed above, for each timestep, we compute the evolution from $s^n$ to $s^{n+1}$ using the 3D image corresponding to timestep $t_n$. The first term $s^1$ of the SSA time series is initialized using the first 3D image, and the subsequent terms are computed using the growth rates deduced from the FE simulations on the 3D image of timestep $t_n$.

This will be specified in the text at the start of the Numerical Modeling Section **Line 140**, alongside a brief presentation of our simulation workflow:

*"The end-goal of our numerical modeling is to simulate the SSA decrease of snow samples over time based on the pore-scale physics, and to compare this decrease to experimental observations. For that, we rely on time-resolved µCT images that were obtained under temperature gradient metamorphism conditions (Pinzer et al., 2012). These µCT scans provide (i) experimental data of the evolution of the SSA over time and (ii) snow-microstructures that can be used for our physical modeling. The computation of a vapor field using a FE simulation, combined with the local curvature of the snow sample, allows us to estimate <$v_nH$> over a given snow microstructure. With Eq. 9, this yields the evolution of the SSA during a given time interval.*

*As we want to reproduce the SSA evolution of entire time series, our general workflow is as follows. For a given experimental time series, we initialize the first term $s^1$ of the simulated SSA values using the SSA deduced from the first µCT image of the experimental time series. Then, the second simulated SSA value $s^2$ is computed by applying the growth rate deduced from a FE simulation performed on the first µCT image. The procedure is then repeated to compute the $n^{th}$ term of the simulated SSA $s^n$ using the already known value $s^{n-1}$ and a FE simulation performed on the n-1$^{th}$ µCT image. This workflow and its different steps are detailed in the Sections below."*

We have also revised the start of Section 4.2 **Line 291** to better explain how the coarse temporal modeling is achieved:

*"In the first step, we compare the temporal evolution of the SSA s between experimental data and the model using a large time step for the modeled data. For that, we downsample the experimental µCT time series to match the coarse temporal resolution and only perform FE simulations on those. Specifically, the modeled SSA values are computed with a coarse time resolution of $\Delta$ = 48h for Series 1 (corresponding to 9 temporal points) and $\Delta$ ~60h for Series 2 (corresponding to 15 temporal points). This reduction in numerical effort allows us to perform a sensitivity study and estimate a value for the effective condensation coefficient α that best matches the experimental data. A fixed constant α is used for each simulation. The range of alpha varies from $10^{-3}$ to 1 for Series 1 and from $10^{-3}$ to $10^{-1}$ for Series 2. For the comparison with these simulated data, we simply use all available experimental SSA data (acquired for a temporal resolution of 8h). The results are shown in Fig.3a,b."*

•selection of appropriate (sub)volumes from the 3D images, pre-processing and volumetric averaging: how exactly are the volumes that feed into the numerical model selected? For several parameters, volumetric averaging is performed (e.g., SSA, curvature, alpha). Is the averaging volume always the same (e.g., the entire domain)? Are we sure to have a size large enough to be representative? For the kinetic coefficient, it is very late in the paper that the concept of "effective coefficient" is introduced. Maybe the concept(s) could be introduced and justified early in an overall context.

For all simulations, we have used the largest available volume. We have performed additional simulations to determine the representativeness of the growth rate computed with different sample sizes. It is shown in the graph below. The volumes used in the article correspond to the largest ones for each sample.

[Figure]

This suggests that the sample sizes used in this study are sufficient to yield representative results in terms of simulated growth rate. We will add this information **Line 179**:

*"The FE meshes of this article are based on the whole available µCT images. We verified that these selected volumes were large enough to yield representative results.. By varying sub-volumes extracted from the center of µCT images at the start and the end of both series ($I(t_1)$, $I(t_{49})$, $\hat{I}(t_1)$ and $\hat{I}(t_{83})$), we found that the simulated growth rate corresponds to a representative value for*

Only the growth rate <$v_n$H> requires averaging. This averaging is performed on the whole triangulated ice-air interface of the sample. The different macroscopic quantities (SSA, growth rate, etc) are computed within the same snow volume to ensure consistency.

No averaging is performed on the condensation coefficient α, as it assumed spatially-constant in the simulations. We will specify early in the text **Line 138** that the use of a spatially-constant alpha is akin to choosing an effective value.

We will specify the use of surface average in **Lines 72** and **127** and we will clarify the definition of the growth rate in **Line 130**:

*"Here the term $\overline{v_n H}$, referred to as the growth rate in this article, is the product of the local interface velocity $v_n$ and the local mean curvature H averaged over the ice-air interface area (the surface average being indicated by an overline over the product)."*

- reason for "numerical (?)" tricks like the "air padding" and iterative (2-times) solution of the energy conservation equation

Adding an artificial air-padding layer around the snow image is a numerical "trick" to obtain a surface mesh with closed outer boundaries using VTK, which is then required for meshing by CGAL. We will clarify the text about the artificial air-padding in **Line 160**:

*"To this end, we employ the open-source Computational Geometry Algorithms Library (CGAL) (The CGAL Project, 2022). Specifically, we use the class Polyhedral_mesh_domain_with_features_3 that implements a tetrahedral mesh of a domain bounded by polyhedral surfaces that are preserved. The provided surfaces need to be closed and free of self-intersections. To obtain such surfaces, we extract the ice-air interface from the binary µCT data (Eq. (11) and (12)) following the procedure from (Krol and Löwe, 2018), namely by applying a Gaussian smoothing and the contour filter from the Visualization Toolkit (VTK) (Schroeder et al., 2006). However, by default this procedure applied to µCT images yields a surface that is open at the boundaries of the domain. In order to obtain closed surfaces, we added a small air-padding (three voxel-thick) around the image. This allowed us to properly define a closed outer boundary suitable for meshing. As detailed below, we provided special care to ensure that the introduction of this artificial air-padding does not perturb the simulation within the snow microstructure itself."*

Solving heat equation twice is made to ensure that the presence of the air-padding does not interfere with the temperature gradient in the snow part of the mesh. We will clarify **Line 181** how we deal with air-padding in the simulation:

*"For the simulation, we need to apply a given temperature gradient across the snow microstructure. However, due to the presence of artificial air-padding, directly applying the required temperature gradient across the whole FE mesh (snow plus air-padding around the image) would result in a smaller temperature gradient within the snow itself (as the air is less conducting than the snow and thus concentrates the temperature gradient). In order to obtain the proper temperature*

3. Error analysis (sections 3.4 / 4.3 / 5.3): While I do like this systematic error analysis, I find the focus very much on "time" related errors; however, I think that other errors (e.g., discretization in space) are at least equally important. These are captured and a bit discussed in section 5.3. Clearer statements – already in earlier sections – would help to better understand the strengths and weaknesses of the approach chosen.

The goal of Section of 3.4 is to propose a framework to analyze how stochastic errors in the growth rate estimation (whatever their origin) propagates and accumulates over time in the SSA. As mentioned later in the manuscript, it is meant to estimate how the temporal resolution and methodological errors combines into a given error for the SSA prediction at a give time horizon. We agree that other types of errors (such as a potential bias due to spatial discretization) are also important.

Regarding the quantification of errors, we have quantified the sensitivity of our results to the mesh resolution. This is shown in the graph below, and will be mentioned in the revised manuscript **Line 170**:

*"We have estimated the sensitivity of our results to the FE mesh. We found that doubling the number of elements in the mesh impacted the growth rate by about 10%. This is small in light of the dependence of the SSA values on the condensation coefficient α investigated in this study. Moreover, the very good agreement between a FE simulation and the analytical solution for a spherical problem (see Sect. 3.6) suggests that our meshing criteria yield an appropriate mesh."*

[Figure]

We also propose to more clearly state at the beginning of Section 3.4 that the stochastic model is meant to quantify how methodological errors accumulate over time in the SSA decrease, and the impact of the temporal resolution in this process. **Line 223**:

*"While the combination of the theoretical solution of the diffusion equation and the SSA evolution is, in principle, exact, the 4D image data processing and the derived SSA are subject to experimental and processing errors. These errors could be of various origins, for instance due to uncertainties related to the estimation of the ice-air interface from the µCT scans or to errors related to the numerical FE discretization. When focusing on the temporal evolution of SSA over time, these errors will accumulate and be propagated into the modeled decrease of s(t). To analyze how these errors translates to the overall SSA decrease, and how this depends on the temporal resolution, we resort to a simple stochastic error treatment."*

Line 7: suggestion: quantify the impact

We will change **Line 7** accordingly:

*"[…], we quantify the impact of these […]"*

Line 16: , normalized per volume. Suggestion: remove the last part of the sentence since you give many examples in the next one.

We will change **Line 16** accordingly:

*"The specific surface area (SSA) of snow is the interface area between ice and air in the microstructure of porous snow, normalized per volume. The SSA is a crucial parameter for [...]"*

Line 20: one key

We will change **Line 20** accordingly:

*"The SSA evolution in time is one key to […]"*

Line 30: well characterized uncertainties

We will change **Line 30** accordingly:

*"[…] computed within well characterized uncertainties due to […]"*

Line 37: maybe you could shortly explain in this paragraph why other effects are (at least here) of second order and neglected, e.g., advective transport, rigid body motion, …

We will mention in this paragraph that other processes could play a role in the evolution of SSA. See the response on Specific Remark 1.

Line 38: one key parameter…

We will change **Line 38** accordingly:

*"In this picture, one key parameter driving snow metamorphism is […]"*

Line 61: I think alpha also impacts numerical effort for the simpler models. Thus, suggestion: Since the choice of alpha has..., it is not surprising that…

Line 64: suggestion: at the expense of …

We will change **Line 61** accordingly:

*"Since the choice of α has a significant impact on numerical effort, it is not surprising that the majority of modeling attempts exist for simplified geometries (mostly spheres) (Adams and Brown, 1982; Colbeck, 1983; Albert and McGilvary, 1992; Miller and Adams, 2009), at the expense of microstructural realism."*

Line 65: detailed is somewhat misleading here (in comparison to the mu-CT models with detailed micro-structure). Suggestion to remove "detailed".

We will change **Line 65** accordingly:

*"[…] are those implemented in snow cover models e.g., […]"*

Line 73: it is not entirely clear to me why not both parameters could be measured experimentally (e.g., extracted from 4D mu-CT) or also both be computed by a model. Maybe you could simply say that what you propose is one possibility.

Furthermore, it seems to me that you also compute the interfacial curvature from the model.

What we meant is that when simulating the SSA evolution from a given 3D image, the computation of the curvature and interface velocity fields are fundamentally different. The first one is purely geometric and is readily accessible with the 3D image. The second is not directly given by the 3D image and requires extra knowledge about the physics at play. It is true that time-series of 3D images could be used to experimentally estimate the interface velocity, but our goal in the paper is to be able to compute how the SSA of a given microstructure evolves over time (without the need for information about the microstructure in the future).

We will specify that in the text in **Line 73**:

*"While the interfacial curvature is a purely geometrical quantity that can directly be computed from a µCT image, $v_n$ is a physical quantity that further depends on the involved physical processes."*

Line 80: later, in the discussion, you (I think, correctly) say that your model works on 3D mu-CT images. I would find it useful to be clear/explicit here in the intro: do you actually use a time-series of 3D images and, for each image, calculate SSA-evolution using the model Or - vs. only use the 1st image and compute from there the SSA evolution for the whole future.

For a 3D image at time $t_n$, we compute the growth rate based on its microstructure and then use this growth rate to update our time series of SSA values from $s^n$ to $s^{n+1}$. We will specify it at the start of the Numerical Modeling section **Line 140**, as detailed in the response to the Specific Remark 2.

Line 115: so far, alpha was called vapor attachment coefficient

We will introduce alpha in **Line 38** defining it "condensation coefficient" and mentioning that other names of alpha appear in the literature:

*"In this picture, one key parameter driving snow metamorphism is the condensation coefficient α, also called attachment, kinetic or sticking coefficient (Libbrecht, 2005; Kaempfer & Plapp, 2009;*

*Krol & Loewe, 2016; Demange et al., 2017; K. Fourteau et al., 2021b; L. Bouvet et al., 2022) that*
*controls the kinetics of vapor deposition and sublimation."*

We will further call alpha "condensation coefficient" in abstract (**Line 8**) and throughout the text.

Line 127: hmm... you also use the mean curvature. Maybe say: this information, together with information about surface curvature, is…

We agree and therefore will change **Line 126** accordingly:

*"[…] this information, together with information about surface curvature, is sufficient to […]"*

Line 128: suggestion (see also comment high up, where SSA is defined): define SSA "per unit volume", remove parenthesis here.

Further down, SSA_V is introduced; maybe this could already be done at the definition high up (or there, one could at least say that the normalization can done in several ways).

minor suggestion: the "evolution equation" for SSA…

We will rewrite this subsection by first introducing both specific surface area definitions and then presenting the surface area evolution equation **Line 125**:

*"In this article, we use of two SSA definitions: specific surface area per unit volume s and specific surface area per ice volume $SSA_v$. They are closely related through the ice volume fraction $\varphi_i$:*

***Eq. 8***

*We mainly work with the quantity s for the rest of the article. However, we note that the quantity $SSA_v$ is more commonly used in the snow community (.e.g Matzl and Schneebeli, 2006), since it directly corresponds to the optical diameter.*

*The solution of heat and mass diffusion equations (Eq. (1)-(3)) with boundary conditions (Eq. (4)-(7)) yields […] As a result, for single grains or statistically homogeneous microstructures, the surface area evolution equation can be expressed as follows: […]"*

Line 130: not entirely clear to me: do you average over the same unit volume as for the SSA? Or is it a smaller average?

The surface average is performed on the ice-air interface of a given snow volume. This strictly corresponds to the interface from which the SSA is defined.

Line 134: simplify: "Eq. (9) allows us…"

We will change **Line 134** removing the word "representation":

*"Equation (9) allows us to […]"*

Line 150: shorten: use mu-CT (after first having introduced the abbreviation) throughout.

Line 150: suggestion: taken instead of extracted

We will change **Line 150** accordingly:

*"The µCT image data were taken from […]"*

We will further change **Line 141**:

*"3.1 µCT time lapse experiments"*

Line 154: shouldn't voxel size be m^3  - or say "in each direction" or "cubic voxel size with side-length"

We will exchange "voxel size" with "voxel side length" in the revised manuscript **Line 154**.

Line 174: I suspect  Gamma here to be the "discretized" surface - correct? This is not entirely consistent with the (general) definition of Gamma at the beginning of 2.2. The same comment holds for H. Think about, if it is worth distinguishing.

Indeed, while Gamma in the "Theoretical background" Section corresponds to the "true" ice-air interface in a snow sample, the Gamma of Line 174 corresponds to the triangulated representation.  We will change Lines 173-175 accordingly:

*"In addition, we computed the boundary weight on each mesh node k of the triangulated ice-air interface $\Gamma_h$*

$$\omega_k = \int_{\Gamma_h} \psi_k \, d\Gamma_h \quad (13)$$

*where [...]"*

Line 180: suggestion: use consistent terminology (e.g., we solve the diffusion equations (1)-(3) - or alternatively, talk about Laplace eqs higher up).

We will introduce the equations in **Line 94** as (stationary) heat and mass diffusion equations mentioning that they can be called Laplace equations:

*"[…], we employ the common assumption of small particle Péclet number (Libbrecht, 2005) and consider stationary heat and mass diffusion equations (i.e. Laplace equations)."*

Line 181: employing the open-source...

We will change **Line 180** accordingly:

*"On the tetrahedral FE mesh with preserved surface, we solve heat and mass diffusion equations (Eq. (1) - (3)) employing the open-source FE software Elmer (Malinen and Råback, 2013)."*

We will further change to the consistent use throughout the whole text**.**

Line 182: these two steps are not entirely clear to me. I suggest to re-formulate it. My questions are in particular:

- why exactly is one solution of the heat equation not sufficient?

- do you really use temperature gradients as boundary conditions (i.e., flux?) and not Dirichlet temperature b.c.'s?

- is the issue the "vertial" boundary, i.e., the air gap or the boundary condition at the top and bottom?

Due to the presence of artificial air-padding, solving the heat equation once on the given FE mesh (snow image plus air-padding around the image) would result in a smaller temperature gradient

within the snow itself. We aim to obtain a correct temperature gradient within the snow structure (the same as in the experiment). To achieve this, we solve the heat equation on the FE mesh twice: the first one tells us how a gradient over the entire domain translates into the snow itself, and the second uses a corrected gradient to match the experimental gradient in the snow itself. We impose Dirichlet boundary conditions on the top and bottom of the FE mesh to apply the macroscopic temperature gradient. Note that we do not impose a microscopic gradient at the boundaries (which would be a Neumann boundary condition).

Following our response to Specific Remark 2, we will reformulate the manuscript **Line 181** to better explain these two steps.

Line 187: use consistent terminology throughout the paper: heat equation vs. diffusion equation, vs. heat diffusion equation (and similar with vapor)

We will change to the consistent use of heat and mass diffusion equations as stated in response to the comment to Line 180.

Line 207: For my clarification: is Calculate Loads simply providing the flux through an interface? If so, maybe simply say so. And, if you like, add: this can be interpreted as deposition and sublimation fluxes. I am not sure if the simulation does provide directly deposition and sublimation information.

The Elmer function Calculate Loads provides the flux at the interface nodes, expressed in kg/s. It then needs to be converted in surface flux expressed in $ks/m^2/s$. We will change **Line 206** accordingly:

*"Finally, the required local interface velocity $v_n$ is computed using the vapor flux deduced from the FE simulation. For this, we use the Calculate Loads option of Elmer that provides the vapor flux $f_k$ (expressed in $kg\ s^{-1}$) at each node $k$ of the ice-air interface. Dividing by the associated boundary weight $\Omega_k$ yields the corresponding deposition/sublimation flux (expressed in $kg\ m^{-2}\ s^{-1}$) over the ice-air interface."*

Line 218: this sentence needs revision (no sense)

We will modify the paragraph for clarity **Line 216**:

*"For that, we use the VTK package and first cut off the small air padding on the sides using vtkClipDataSet. Then, the triangulated ice-air interface is extracted. The local interface velocity $v_n$ is directly taken from the FE simulation using Eq.(17) . For the local curvature H, we employ the image analysis derived in Krol and Loewe (2018), based on the shape operator, as explained in Section 3.2. Finally, the surface integration for the average in $<v_n H(t_n)>$ takes into account the variable element size of the triangular mesh of the ice-air interface."*

Line 222: During a 1st read, I had the impression that the error discussion focusses on discretization errors in time.

I would find it useful to clearly state the role of the variance sigma (maybe after having said "of unknown origin") and how it can include dicretization in space or smoothing related errors. Maybe even say already here how one may estimate sigma.

We will specify in the text that these errors could be of whatever origin, for instance due to the variability between the actual ice-air interface, and the reconstructed surface through μCT. **Line 223**:

*"While the combination of the theoretical solution of the diffusion equation and the SSA evolution is, in principle, exact, the 4D image data processing and the derived SSA are subject to experimental and processing errors. These errors could be of various origins, for instance due to uncertainties related to the estimation of the ice-air interface from the µCT scans or to errors related to the numerical FE discretization. When focusing on the temporal evolution of SSA over time, these errors will accumulate and be propagated into the modeled decrease of s(t). To analyze how these errors translates to the overall SSA decrease, and how this depends on the temporal resolution, we resort to a simple stochastic error treatment."*

**Line 223: equations (heat and mass)**

We will change to the consistent use of heat and mass diffusion equations as stated in response to the comment to Line 180.

**Line 225: assess (instead of address)?**

We will change **Line 225** accordingly:

*"[…] error treatment to assess the impact of these errors."*

**Line 227: is there a reason to use t' here and not tau anymore? If not, I suggest to unify the notation**

There is no specific reason for using t'. We will exchange t' with $\tau$ in **Eqs. (19), (20), (22) and Lines 228 and 231**. We will further correct **Eq. (21)** using $\tau$ for all terms in the equation:

*"$r(\tau) = r^{true}(\tau) + \delta r(\tau)$"*

**Line 228: do we really want to introduced "decay rate"? is it not rather "rate of evolution"**

We will change **Lines 72, 130, 134, 216, 228, 231, 363 and 442** determining $v_n H$ as "growth rate" consistently throughout the text.

**Line 233: its rather "computations" than "measurements" that are affected, I think**

Indeed. We will change **Line 233** accordingly:

*"[…], which affects the computations at each time step."*

**Line 254: should this be "and" and not "as" (?)**

We will clarify the sentence **Line 253**:

*"We set up a complex numerical workflow that starts from a voxel image, computes the interface velocity $v_n$ from a FE simulation, and eventually yields the growth rate $<v_n H>$ after surface integration."*

**Line 257: should we say that it is an isothermal situation?**

The vapor problem between the outer and inner shells is actually temperature-independent, rather than isothermal. There is no need for a temperature field in this case. We will revise the paragraph and explain that the problem is temperature-independent **Line 255**:

Figure 1. b) would we not expect v_n to be either positive or negative everywhere?  i.e., why is the scale going from negative to positive values?

From a physical point of view, we expect the vapor field to deposit on the inner sphere and water vapor to be added from the outer sphere to maintain the mass balance. In terms of interface velocity $v_n$, it is positive on the inner sphere (vapor deposition) and negative on the outer sphere (vapor sublimation). We will expand the **caption of Figure 1** specifying that there is sublimation and deposition:

*"Figure 1. [...]  b) Clip of the outer and inner spherical shells with visible elements colored by the interface velocity $v_n$ (sublimation in blue, deposition in red). [...]"*

We will also extent the text **Line 270** to better explain what Fig. 1b illustrates:

*"After solving the vapor equation, with appropriate boundary conditions, we obtain the interface velocity $v_n$, shown in Fig.1b. As expected, we observe a positive velocity on the inner shell, corresponding to vapor deposition, and a negative velocity on the outer shell, corresponding to sublimation."*

Line 287: please introduce ancronymes when first used.

We will change **Lines 286-287** introducing the acronym:

*"[...] showing the best root mean square error (RMSE) agreement [...]"*

We will further change **caption of Figure 1c** and **Line 303** using only the acronym.

Line 296: does this mean that you use e.g., the 1st image to compute SSA evolution form time 0 to time 48, then the image from time 48 for the next 48h, and so on? See also my remark in the intro-section. I would find it helpful to be more clear / explicit, either above or here in the numerics section.

Yes it is exactly that. We compute the growth rate at a given time $t_n$ using the corresponding microstructure and apply it to the SSA to compute its value at time $t_{n+1}$. We will clarify this in the text, following our response to the Specific Remark 2.

Figure 4: a color scheme that also works for people with color deficiency would be appreciated. Alternatively, use different "dots" or line-styles

We will improve readability of **Figure 4** by changing a color and using dashed lines:

[Figure]

We will change **Line 308** accordingly:

*"[…] for Series 2 is lower despite higher data scattering."*

Indeed, the stochastic model not only incorporates the temporal resolution but also the level of methodological errors. The resulting error on the SSA is the interplay of both. However, the Section mainly focuses on how varying the temporal resolution (while assuming a constant level of methodological errors) impact the modeled SSA decrease. Therefore, we would prefer to keep the title of the Section as such.

It is true that once the stochastic model is adjusted to the simulated data, we obtain a variance that *in principle* characterizes the level of stochastic methodological errors. However, before further interpreting this adjusted variance value, it would likely be beneficial to quantify the robustness of the method and to ensure that this adjusted value is indeed indicative of the overall methodological error in the simulation workflow. This is an interesting prospect for future work.

We will change **Line 310** accordingly:

*"[…] we performed simulations with a time step refined down to the time interval between […]."*

We will change **Line 321** accordingly:

*"On the one hand, this […]"*

Line 367: suggestion: "nevertheless" instead of "as a result of the numerical effort"

We will change **Line 367** accordingly:

*"Nevertheless, we were able to [...]"*

Line 408: Indeed, I think it is a remarkable result that a somehow volumetrically averaged kinetic coefficient seems to be sufficient to explain SSA evolution. The message is here, but not very "pointy".

We will draw attention to this result by adding the following sentence to **Line 405**:

*"It is quite remarkable that despite variations of the condensation coefficient at the micro-scale, their collective behavior can be appropriately described through the use of a single α value. Indeed, in principle, the assumption of a constant α [...]"*

Line 420: how come the volume for the simulation impacts the experimental parameters? Can this be reformulated?

If the volume are quite close or smaller than the REV size, one could observe fluctuations in the SSA values that are due to changes in the observed Region Of Interest rather than variations due to actual evolution of the macroscopic SSA. This will be reformulated in the text **Line 418**:

*"Second, the volume of interest considered here for the simulations is relatively small, in particular for Series 2. This might lead to some non-representativeness issues and fluctuations in the measured SSA. This could explain the noisy nature of the experimental parameter curves."*

Line 425: yes! I think, you should not entitle section 3 as "temporal" only and have the discussion along the line of this nice summary.

Following the Specific Remark 3, we will mention earlier in the manuscript **Line 223** that the stochastic model incorporates the interplay between temporal resolution and methodological uncertainties.

Line 434: I am not sure that non-convergence of the solver is linked to this simplification. I would keep the two discussions separate. You could e.g., justify (higher up, where it is first said) to neglect latend heat for simplicity with the argument to increase numerical stability. Then, in a separate paragraph under 5.4. say that numerical improvements might help to increase numerical convergence.

This simplification is a trade-off between numerical simplicity and physical realism. It helps the convergence of the solver, as the numerical problems to be solved are smaller and less complex, but remove a potentially important physical process.

We will explain early in the revised manuscript why latent heat is neglected **Line 108**:

*"As by Krol and Löwe (2016), the latent heat during the sublimation and deposition is neglected for reduced model complexity."*

We will clarify why this simplification helps convergence, but that despite it there are still converge issues, and propose so remedy for it **Line 432**:

*"This leads to a slightly simpler numerical situation where heat and vapor are coupled only one way, and the heat diffusion equation can be solved in advance. This strategy reduces the numerical cost of the method and facilitate the convergence of iterative solver used in the FE software. Despite this simplification, we still observe that the vapor solver had issues to converge for a few microstructures, which explains a few missing points in the modeled time series (e.g., Fig. 4. The convergence of the FE simulations depends on the employed mesh and on the value of α. It could be facilitated by improving the mesh quality or increasing the maximum number of iterations."*

Line 436: It seems not correct that latent heat can contribute to mass fluxes. It might possibly impact them. Please reformulate.

We will rephrase the sentence in **Line 435**:

*"While this one-way coupling assumption eases the numerics, it was previously shown Fourteau et al. (2021) that for low density or fast kinetics, latent heat significantly contributes to the heat fluxes in snow and may thus likewise impact the volume averaged rate term $v_n H$."*

Line 446: I think this such a central result / conclusion that it is worth cross-checking consistency with terminology and already higher up in the document introduce the "effective kinetic coefficient", I think it is to be understood as a volumetric average.

We will introduce the notion of an "effective condensation coefficient" early in the revised manuscript in Sect. 2, **Line 123**:

[revised manuscript text omitted]

---

## Author Response (AR2)

Dear Editor

Please find the updated version of the paper. This document contains our responses to Dr. Kaempfer's comments and the marked-up version of the paper. As proposed by Dr. Kaempfer, we have included the schematic figure describing the workflow. We have also corrected recently detected typos and listed them after our responses to Dr. Kaempfer's comments.

Kind regards

Anna Braun on behalf of all co-authors

**Responses to referee report from Thomas Kaempfer**

We are thankful to Thomas Kaempfer for the valuable suggestions for improving the manuscript.

We have copied the comments from Thomas Kaempfer in blue. Our corresponding responses are available below each comment in black. The bold line numbers correspond to the updated version of the manuscript.

Some further typos were detected during the last reading and are briefly listed after the responses to the comments.

Kind regards

Anna Braun on behalf of all co-authors

I find this figure very illustrative and it camptures the workflow nicely and in a compact form. Would there be space to include it in the paper?

We have included the schematic figure describing the worklow. The figure is introduced in the beginning of Section 3, **Lines 164-165**:

*"The workflow and its different steps are detailed in the sections below and illustrated in Fig. 1."*

[Figure]

*"**Figure 1.** Schematic illustration of the workflow used in this study in order to compute modeled SSA values $s^n, n = 2, ..., N+1$. On each 3D image of a µCT time-lapse series, a tetrahedral mesh is produced and a heat and mass diffusion simulation is conducted. The simulated interface growth velocity $v_n$ is displayed in color in the Figure (blue corresponding to a receding interface and red to a growing interface). In the post-processing step, the growth rates $\overline{v_n H}(t_n)$ are extracted and used to model the SSA evolution according to Eq. 18."*

**Line 158: an FE simulation**

We have changed **Line 158** accordingly:

*"[…] an FE simulation […]"*

The same error was corrected in **Lines 162, 164, 201 and 297**.

**Line 135: we use two…**

We have changed **Line 135** accordingly:

*"[…] we use two […]"*

"Second, the volume of interest considered here for the simulations is relatively small, in particular for Series 2. This might lead to some non-representativeness issues and fluctuations in the measured SSA. This could explain the noisy nature of the experimental parameter curves.":

and yet the volume seems to be far large enough w.r. to the computed parameters (see your remark / clarification higher up). This is somewhat counter-intuitive.

I have no nice, short explanation for this; so probably best leave your text as it is.

From what we understand, there is no strong contradiction between the fluctuations of Series 2 and our REV analysis. The fluctuations of Series 2 are of a few percents only, which is small enough to still consider each individual point to be fairly representative. However, the fluctuations in Series 1 appear even smaller. We thought that this is an interesting observation in regards to the variance of the data and postulated that this might be explained by the sample size difference between the two series.

We have reworded the text in **Lines 470-473** to emphasize that the smaller size of Series 2 samples only leads to relatively small fluctuations:

"Second, the volumes of interest considered here could be larger, in particular for Series 2. This size might lead to some non-representativeness issues and small fluctuations in the measured SSA. This could explain the slightly noisy nature of the experimental parameter curves in Series 2 compared to Series 1."

**Line 487: and facilitates the convergence of the iterative ...**

We have changed **Line 487** accordingly:

*"[…] and facilitates the convergence of the iterative […]"*

**Line 489: close parenthesis**

This error only appears in the author's response to Thomas Kaempfer's review. The manuscript does not contain the error.

**Line 491: parentesis missing for citation**

This error is also only found in the author's response to Thomas Kaempfer's review and not in the manuscript.

Moreover, typos have been detected and corrected in **Lines 5, 30, 58, 69, 94, 128, 169, 171, 180, 191, 213, 255, 269, 380, 430, 477, 491 and the caption of Fig. 4**.

[revised manuscript text omitted]